# Fast and Fluent Diffusion Language Models via Convolutional Decoding and Rejective Fine-tuning

**Yeongbin Seo**    **Dongha Lee**    **Jaehyung Kim**    **Jinyoung Yeo** *

Department of Artificial Intelligence
Yonsei University
{suhcrates,donalee,jaehyungk,jinyeo}@yonsei.ac.kr

## Abstract

Autoregressive (AR) language models generate text one token at a time, which limits their inference speed. Diffusion-based language models offer a promising alternative, as they can decode multiple tokens in parallel. However, we identify a key bottleneck in current diffusion LMs: the **long decoding-window problem**, where tokens generated far from the input context often become irrelevant or repetitive. Previous solutions like semi-autoregressive address this issue by splitting windows into blocks (sacrificing bidirectionality), but we find that this also leads to **time-interval expansion problem**, sacrificing the speed. Therefore, semi-AR eliminates the main advantages of diffusion models. To overcome this, we propose Convolutional decoding (*Conv*), a normalization-based method that narrows the decoding window without hard segmentation, leading to better fluency and flexibility. Additionally, we introduce Rejecting Rule-based Fine-Tuning (R2FT), a post-hoc training scheme that better aligns tokens at positions far from context. Our methods achieve state-of-the-art results on open-ended generation benchmarks (e.g., AlpacaEval) among diffusion LM baselines, with significantly lower step size than previous works, demonstrating both speed and quality improvements. The code is available online (https://github.com/ybseo-ac/Conv).

## 1 Introduction

Autoregressive (AR) language models (LMs) have shown remarkable performance in recent years [31, 32, 6, 1]. Diffusion-based LMs are gaining attention [3, 36, 29, 2] as they offer potential solutions to two fundamental limitations of ARLM: (1) ARLMs must generate text one token at a time, which imposes an inherent limit on speedup. In contrast, diffusion models can be further accelerated by reducing decoding steps while maintaining the same output length. (2) ARLMs are inherently unidirectional, relying only on past context, whereas diffusion LMs can leverage bidirectional context [49]. Since human language understanding involve both past and future context, bidirectionality offers a more flexible and human-like approach.

However, we find a critical inherent drawback of diffusion LLMs, called **long decoding-window (LDW for simplicity) problem**, which is a key bottleneck in fluent text generation. While AR models focus on predicting a single token that is directly attached to the previous context, diffusion LMs treat all positions as potential targets for decoding. Here, the problem occurs: tokens suggested at positions far from the previous context tend to be less aligned and more random. As a result, naive categorical sampling during decoding often yields outputs that are poorly grounded in the given context.

Several approaches have been proposed to address this issue, including semi-AR (e.g., LLADA [29], Block-diffusion [2]). Semi-AR divides the generation window into multiple blocks and decodes them sequentially, therefore shortening the decoding area. This suppresses decoding of positions far from

---

*Corresponding author

previous context, directly mitigating the LDW problem. It is also reported to improve performance in downstream tasks [29].

However, we analyze that semi-AR still has significant inherent limitations. (1) Most notably, it sacrifices decoding speed. Reducing the step size in semi-AR is challenging, as dividing the total steps across multiple blocks leads to a rapid degradation in overall text generation quality, due to an issue we term the **time-interval expansion problem**. (2) Also, semi-AR inevitably sacrifices bidirectionality. As semi-AR lacks both the speed advantage and bidirectionality, it removes most of the motivation to choose diffusion LMs over AR models.

Therefore, we propose two approaches that both address the LDW problem while retaining speed and flexibility. (1) **Convolutional decoding (*Conv*)**: This narrows the decoding window similarly to semi-AR, but uses normalization instead of strictly splitting it into blocks, bypassing quality degradation. (2) **Rejecting Rule-based negative Fine-Tuning (R2FT)**: A post-hoc training scheme that aligns distant tokens with previous context, removing the need for narrowing the window. We empirically demonstrate that combining these methods leads to more fluent and coherent open-ended responses.

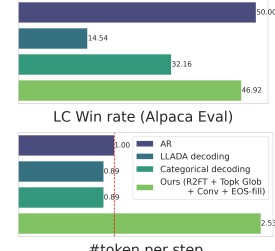

Figure 1: G-eval score and sampling speed on AlpacaEval. Ours achieves SOTA.

For the experiments, we focus on testing an instruction-guided open-ended answer generation ability. Previous studies primarily evaluated diffusion LMs using natural language understanding (NLU) benchmarks that are measured with multiple-choice metrics or string matching metrics [29]. However, this makes it difficult to assess the biggest challenge of diffusion LMs: fluent open-ended text generation. We observe that previous diffusion LMs [29] which showed remarkable scores on NLU tasks (e.g., MMLU [16]) perform poorly on benchmarks to evaluate open-ended answer generation such as AlpacaEval [44], often generating broken sentences. Therefore, we use benchmarks like AlpacaEval and the metric of G-Eval [26], which is known as most aligning with human evaluation, showing that our method achieves state-of-the-art performance among diffusion-LM baselines, with even around one-third the step size of previous work Figure 1. For bidirectional generation, we provide a proof of concept through sampling patterns (Figure 7). Since most LM tasks assume a unidirectional scenario, benchmark experiments are deferred to future work.

Our novel contributions are summarized as follows:

- We define the long decoding-window problem, one of the core challenges in diffusion LMs.
- We identify and analyze the inherent speed limitation of semi-AR (i.e., block diffusion), termed the time-interval expansion problem, which was unrecognized in prior studies.
- We propose two alternative methods to address the long decoding-window problem without sacrificing speed or bidirectionality, *Conv* and R2FT.
- We demonstrate that models equipped with our methods achieve state-of-the-art performance among diffusion LM baselines, on open-ended answer generation tasks.

## 2 Preliminaries and motivation

### 2.1 Diffusion LM

In the image generation domain, a diffusion model refers to a model trained through a forward process that adds gaussian noise for $T$ steps to the original data ($\mathbf{x}_0$), and a reverse process that learns to remove the noise, gradually denoising from pure noise ($\mathbf{x}_T$) through intermediate steps ($\mathbf{x}_t, ..., \mathbf{x}_1, \mathbf{x}_0$) [17, 40, 42, 10, 41]. The forward process is formulated as $q(\mathbf{x}_t|\mathbf{x}_0) = \mathcal{N}(\mathbf{x}_t; \sqrt{\overline{\alpha}_t}\mathbf{x}_0, (1 - \overline{\alpha}_t)\mathbf{I})$ where $\overline{\alpha}_t = \prod_{i=1}^{t} \alpha_i$, $\alpha_t \in [1, 0]$. The reverse process is training to minimize Negative ELBO (NELBO):

$$\mathcal{L}_{\text{NELBO}} = \mathbb{E}_q \left[ \underbrace{-\log p_\theta(\mathbf{x}_0|\mathbf{x}_1)}_{\mathcal{L}_{\text{recons}}} + \underbrace{\sum_{t=2}^{T} D_{\text{KL}}[q(\mathbf{x}_{t-1}|\mathbf{x}_t, \mathbf{x}_0) \| p_\theta(\mathbf{x}_{t-1}|\mathbf{x}_t)]}_{\mathcal{L}_{\text{diffusion}}} \right] + \underbrace{D_{\text{KL}}[q(\mathbf{x}_T|\mathbf{x}_0) \| p_\theta(\mathbf{x}_T)]}_{\mathcal{L}_{\text{prior}}} \quad (1)$$

As the forward process is designed for continuous data (e.g., image), applying this framework to discrete data (e.g., text) has been a major bottleneck in the study of diffusion LMs. However, the

remarkable work of D3PM by Austin et al. [3] opened the door by defining the forward process as follows:

$$q(\mathbf{x}_t \mid \mathbf{x}_0) = \text{Cat}(\mathbf{x}_t; \overline{\mathbf{Q}}_t \mathbf{x}_0), \quad \overline{\mathbf{Q}}_t = \mathbf{Q}_t ... \mathbf{Q}_2 \mathbf{Q}_1 \tag{2}$$

Where $\text{Cat}(\mathbf{x}; p)$ is a categorical distribution over the one-hot row vector $\mathbf{x}$ (dimension of vocab size) with probabilities given by the row vector $p$, and $\mathbf{Q}_t$ is a transition matrix at timestep $t$. This formulation laid the foundation for many subsequent studies to follow [19, 15, 7, 27, 34, 43, 22, 48, 38]. Furthermore, Sahoo et al. [36] proves that the NELBO objective of D3PM can be extremely simplified to Equation 3 by following certain assumptions:

$$\mathcal{L}_{\text{NELBO}}^{\infty} = \mathbb{E}_q \int_{t=0}^{t=1} \frac{\alpha_t - \alpha_{t-dt}}{1 - \alpha_t} \log \langle \mathbf{x}_\theta(\mathbf{x}_t, t), \mathbf{x}_0 \rangle dt, \quad t \in [0, 1], \; \alpha_t \in [1, 0] \tag{3}$$

This objective function is the same as masked language model (MLM) such as BERT [9], while the only difference is the forward process with a more complex masking pattern. The assumptions are as follows: (1) The time space is continuous, which indicates $T \to \infty$, $dt \to 0$. (2) The fully noised state $x_T$ means all tokens are masked, and the reverse process is unmasking them. Therefore, diffusion LM is trained to predict tokens at masked positions, and during decoding, it unmask each position of a fixed-size decoding window over $S$ steps ($S \approx T$).

This formulation demonstrates the most stable convergence of the validation loss compared to earlier approaches (e.g., SEDD and D3PM) [36]. Several concurrent works (LLADA, Block-diffusion, MD4 [38]) reach the same formulation by intuition and empirical study, establishing it as the current consensus in the field. In this work, following Sahoo et al. [36], we refer to diffusion LMs trained with this objective as **MDLM** (masked diffusion language models). (See §F for a more detailed review of related works.)

**Notation of $\pi$ and $p$** In this work, two kinds of conditional probability are defined. The conditional probability defined with the $\pi$ term, $\pi(x_i | x_{-i})$, is the probability of a token at position $i$, conditioning on the tokens provided at positions $-i$. On the other hand, the conditional probability defined with the $p$ or $q$ term, $p(x_t | x_{t-1})$, refers to the probability of a token at a given position (e.g., $i$, usually not stated) and the timestep $t$, conditioned on the token of the same position at timestep $t - 1$.

## 2.2 Bottleneck of MDLM: long decoding-window problem

We analyze that the main bottleneck of MDLMs in open-ended text generation lies in the overly broad decoding candidate space, called **long decoding-window (LDW) problem**. Unlike AR models, which decode one token at a time, MDLMs predefine a decoding window with a fixed window size (denoted as $L$, fixed to 1024 in our work) and treat all $L$ positions as candidates for decoding at every step. However, if the position is farther from the previous context (equal to the prompt if the first step), the model tends to predict more context-irrelevant tokens. In our analysis, the pattern of irrelevance can be summarized as a combination of "high-prior tokens" and "repetition of the previous context".

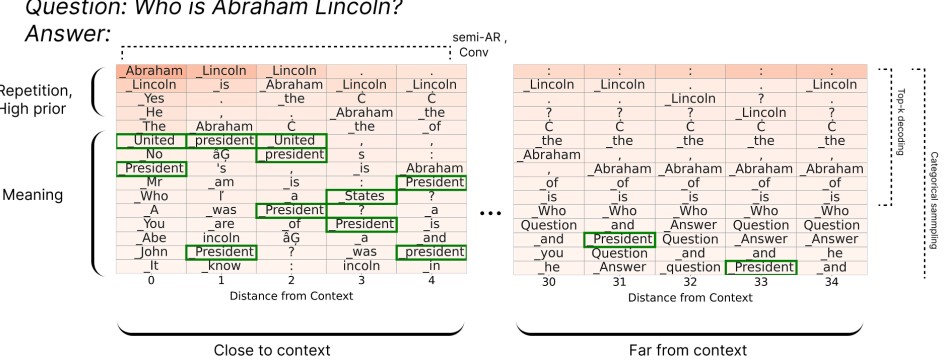

Figure 2: Candidate zone of the first inference step given the previous context "Q: Who is Abraham Lincoln? A:". The left box shows positions close to the context (0 to 4), and the right box shows positions relatively far from the context (25 to 29). Darker red indicates higher confidence of the model. Informative token for given question (e.g., "president", "United States" ) are **outlined**.

An example of a candidate zone is presented as a table in Figure 2 (with the x-axis indicating the unmasked position which is distance from the previous context, and the y-axis showing the top-$k$ ranks for each position). MDLM infers this table at every step, and samples tokens from it.

We observe that the candidate zone is typically composed of four types of tokens clustered like a layer (*repetition*, *high-prior*, *meaning*, *noise*), often appearing sequentially from the top rank, as illustrated in Figure 2 and Figure 11. (1) **Repetition**: A zone where the model repeats tokens from the previous context. In Figure 2, we observe excessive repetition of tokens from the previous context (e.g., "Question", "Answer", ":"), which is not helpful in the current step. This phenomenon likely stems from the inherent tendency of LLMs to repeat previous context, which has also been reported in AR models when generating longer responses [18, 46]. (2) **High-prior**: A zone composed of tokens with high term-frequency, which are often functional words (e.g., ".", "the", "is"). This zone tends to appear thicker when farther from the previous context. We analyze that, from the Bayes-form decomposition $\pi(x_i|x_{\neq i}) \propto \pi(x_{\neq i}|x_i)\pi(x_i)$, if the likelihood term (indicating the relevance with context) weakens, the prior (proxy for term-frequency) dominates. (3) **Meaning**: Tokens containing meaningful information aligned with the previous context (e.g., "president", "United States"). This zone becomes more prominent when it is closer to the previous context. (4) **Noise**: Tokens that are unrelated to the previous context, filling up the rest of the lower rank.

The key property is that, as the distance from the previous context gets farther, the *meaning* zone tends to fall below the *repetition* and *high-prior* zones. This trend is also consistent with statistical observation (Figure 3). We measure the sum of probabilities of *high-prior* (top-100) and *repetition* (tokens overlapping the prompt) tokens at varying distances from the instruction prompt. For both, the probabilities are lowest at distance 0, indicating weak dominance. However, they peak after around 5-10 positions. The *high-prior* continues to occupy around 50% throughout the window.

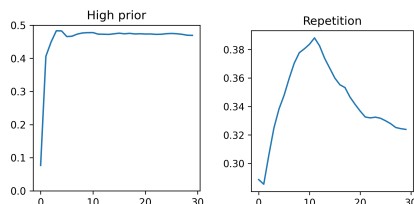

Figure 3: X-axis is distance from instruction prompt, Y-axis is summed probability of *high-prior* and *repetition*.

This property leads to the following problems. If sampling is performed over the entire decoding window, deterministic decoding becomes difficult. In other words, if tokens are sampled from top-ranked candidates, there is a high chance of selecting only *repetition* or *high-prior* tokens in positions far from the previous context, leading to generating repetitive and dull text (Figure 11).

The tendency to generate repetitive and dull text under highly deterministic decoding (e.g., greedy decoding) is also observed in ARLMs [18]. However, AR has the advantage of sampling from a very short decoding window (size of 1) that is closest to the previous context, where *meaning* tokens gain more preference. As a result, this issue can often be effectively addressed simply by using slightly more probabilistic yet deterministic decoding strategies, such as nucleus or top-$k$ sampling [46].

**Formulation**    We provide a theoretical explanation of the LDW problem. Consider a scenario in which the model decodes $x_i$ at the first sampling step, where $x_i$ denotes a mask token that is in the $i$-th position from the given context. AR decoding employs a fixed value $i = 1$, whereas in a diffusion LM with LDW, we may assume $i = 20$ or higher values. The probability of candidate $c$ for $x_{21}$ is $\pi(c|\tau) = \mathbb{E}_\tau[\pi(\tau|c)\pi(c)]$, where $\tau \in T$ is all possible combination of $\{x1, x2, ..., x20\}$. The cardinality of $T$ is $|V|^{20}$ (V is vocab size), infinity. Therefore, posterior $\pi(c \mid \tau)$ converges to $\pi(c)$.

$\pi(c)$ represents the probability of a candidate that can be easily predicted without any contextual information. Typical examples include high-prior tokens (e.g., function words) or simple repetition of given context. This applies regardless of whether $i = 20$, 21, or 22. Consequently, as illustrated in the upper part of Figure 11, top candidates often include patterns such as "Question: Question: Question:" (repetition of given context) or "the the the" (high-prior tokens).

## 3    Existing solutions for LDW problem and their limitations

### 3.1    Categorical sampling

Sahoo et al. [36], who introduced MDLM, addressed this problem using categorical sampling. This method randomly samples from the full vocabulary instead of focusing on the top-ranked tokens. It is expressed as Equation 4, where $a_t$ is a noise schedule, and $s = t - dt$. Both $\mathbf{m}$ and $\mathbf{x}_\theta(\cdot)$ are $V + 1$ dimensional probability distributions, where $\mathbf{m}$ assigns probability 1 to the mask index $(V + 1)$, and

$\mathbf{x}_\theta(\cdot)$ is a softmax distribution inferred by the model based on the state $\mathbf{x}_t$.

$$p_\theta(\mathbf{x}_s|\mathbf{x}_t) = q(\mathbf{x}_s|\mathbf{x}_t, \mathbf{x} = \mathbf{x}_\theta(\mathbf{x}_t)) = \begin{cases} \mathrm{Cat}(\mathbf{x}_s; \mathbf{x}_t), & \mathbf{x}_t \neq \mathbf{m}, \\ \mathrm{Cat}\left(\mathbf{x}_s; \mathbf{m} + \frac{dt}{1-\alpha_t} \cdot \mathbf{x}_\theta(\mathbf{x}_t)\right) & \mathbf{x}_t = \mathbf{m}, \end{cases} \quad (4)$$

This setup allows both high- and low-ranked tokens to remain as candidates, reducing the chance of sampling from *repetition* or *high-prior* tokens that dominate the top ranks. In turn, it also increases the likelihood of bypassing the meaning zone, jumping directly to the noise zone. For this reason, text generated by categorical sampling often appears grammatically fluent but less aligned (Figure 15).

## 3.2 Semi-AR: time-interval expansion problem

Recent studies (LLADA [29] and Block diffusion [2]) systemized semi-AR decoding to successfully address the LDW problem. Although it was not explicitly mentioned, we analyze that the performance improvements of those methods were the result of addressing this problem.

This approach divides the decoding window into smaller blocks and decodes them sequentially, which restricts sampling from positions far from the context. As a result, it directly bypasses the LDW problem. Nie et al. [29] reports that applying semi-AR decoding raises downstream task performance to a level comparable with AR models, while the performance is poor without it.

However, semi-AR has two critical drawbacks that undermine the motivation for using diffusion LMs. It sacrifices decoding speed and directionality.

**Semi-AR sacrifices decoding speed** We found that though the total step size is maintained, splitting the decoding window and step size into multiple blocks makes text quality drop. Therefore, when using semi-AR, the total step size must be kept high to maintain the quality of the overall output, which removes the speed advantage.

This observation contradicts intuition, as it is more natural to expect that if the sample size per step remains, the text quality should also be maintained. For example, decoding a window of size 1024 in 128 steps, or splitting it into two 512-token blocks and decoding each in 64 steps, both result in an average of 8 tokens sampled per step. Thus, one might expect the sampling quality to be similar in both cases. However, in practice, the quality drastically degrades (Figure 4). We term this phenomenon **time-interval expansion problem.**

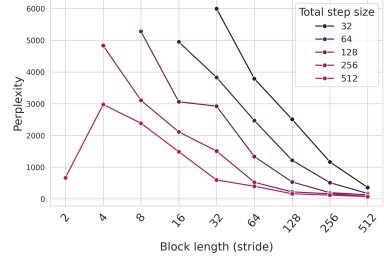

The rationale behind is as follows. Let the decoding window size $L$, the total step size $S$, and the number of blocks $b$. Then, the window size and step size per block are denoted as $L_b = L/b$ and $S_b = S/b$, respectively. If standard decoding of diffusion LMs is a single reverse process with time span $t \in [1, 0]$, then semi-AR performs $b$ multiple reverse processes, each also with time span $t \in [1, 0]$. When the infinitesimal time interval is $dt = 1/S$, the time interval within each block becomes $dt_b = 1/S_b$, which is $b$ times larger than $dt$. As $b$ approaches $S$, $dt_b$ approaches 1, which is no longer infinitesimal.

Figure 4: Perplexity (y-axis) of the text samples from pretrained MDLM, applying **semi-AR** decoding with different **block sizes** (x-axis) on fixed $L = 1024$. Each line corresponds to a fixed $S$.

This leads to a violation of the continuous-time assumption in the objective of MDLM (§2). The training objective of MDLM is simplified thanks to assuming a continuously divisible time span ($dt \to 0$). However, $dt_b$ is $b$ times larger than $dt$, leading to a mismatch between training and inference. For this reason, text quality depends more on $dt_b$ than on $L/S$, as observed in Figure 4. To state it more intuitively, when $L_b$ and $S_b$ get smaller, the model has less freedom to choose an unmasked position, which seems to cause the drop in performance. (We provide another version of theoretical explanation based on the hazard function in §G.1)

Then, how did the previous works address this? LLADA [29] and Block diffusion [2] also share the same training objective and assumption as MDLM and are likely to face the same problem. However, they avoid facing the problem by just setting very high $S$ as default ($S$ as 512-1024 and $L$ as 1024).

It may seem natural to assume that decoding with a window size 1024 with a total step size 1024 results in a "1 token per step" manner, giving a speed comparable to AR. However, this is a misconception. In practice, the actual content in the decoding output typically consists of only around 200–300

tokens on average, with the rest being padded by EOS (i.e., end of segment) tokens. Therefore, in reality, diffusion LMs with $S = 1024$ and $L = 1024$ actually generate 300 tokens with 1024 steps, which is over three times slower than AR. This implies that, to claim a speed advantage, diffusion LMs must reduce the total number of steps $S$ to a substantially smaller fraction (e.g., one-third) of the decoding window size $L$. However, such a reduction causes a severe degradation in generation quality for semi-AR as discussed. Consequently, it is inherently difficult for semi-AR approaches to claim a speed advantage.

**Semi-AR sacrifices bidirectionality**   Since semi-AR assigns blocks sequentially from the left, regardless of whether the input context is on the left or right, it loses its bidirectional decoding ability. We provide a detailed explanation for this in §H.1, and note that bidirectional generation tasks are not included in this paper.

In §1, we highlighted the key advantages of diffusion LMs as speed and bidirectionality. However, semi-AR sacrifices both, making it a poor alternative for AR in practice.

# 4   Proposed methods

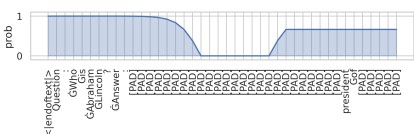

In this section, we propose two approaches that address the LDW problem without suffering from the time-interval expansion, therefore preserving the sampling speed and also bidirectionality.

Figure 5: Convolution normalizer ($s_i$) for each position across decoding window, given bidirectional context.

## 4.1   Convolutional decoding

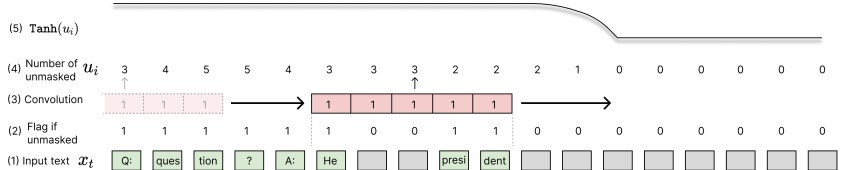

Figure 6: Pipeline of convolutional decoding, from (1) to (5). Gray boxes are mask tokens.

We introduce **convolutional decoding (*Conv*)**, a method that narrows the decoding window as in semi-AR, but achieves this via a normalization mechanism instead of hard segmentation. This helps mitigate the time-interval expansion problem. *Conv* is an alternative to semi-AR with the following advantages: (1) It allows a more flexible adjustment of the decoding window than semi-AR, leading to improved performance. (2) It is simpler to implement. (3) It preserves bidirectionality. We provide more clarification about bidirectionality in §H.1.

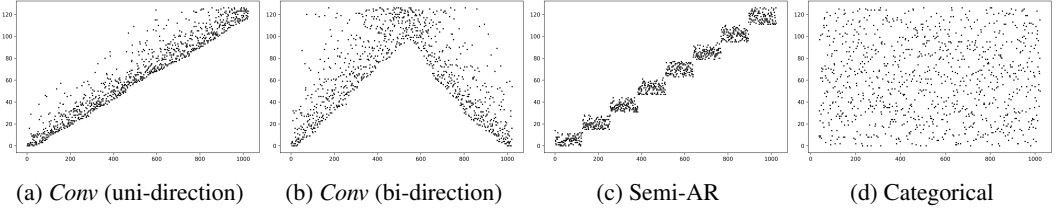

(a) *Conv* (uni-direction)      (b) *Conv* (bi-direction)      (c) Semi-AR      (d) Categorical

Figure 7: Unmask positions (Y-axis) over decoding steps (X-axis) for different decoding strategies. $L = 1024$, $S = 128$, with a kernel size and block size of 128. All strategies begin with a BOS (begin-of-sentence) token at the first position. In (b), a BOS token is also placed at the last position.

The proposed method is as follows. Let $p(x_{ij})$ denote the probability of a token at top-$j$ rank of $i$-th position within the decoding window inferred by the model. The probability after applying the convolutional transformation is $p^{Conv}(x_{ij}) = p(x_{ij}) \cdot s_i \cdot s_{norm}$, where the transformation function for each position is $s_i = g(u_i)$. As illustrated in Figure 6, $u_i$ is the number of unmasked positions within a fixed distance (i.e., **kernel size**) around position $i$, computed using a 1D convolution

filter. $g(u_i)$ is a function of $u_i$, and we find that `tanh` performs best. The normalizing constant is $s_{norm} = \frac{\sum_i^L \sum_j^V p(x_{ij})}{\sum_i^L \sum_j^k p(x_{ij}) \cdot s_i}$, which ensures that $\sum_j p^{Conv}(x_{ij}) = 1$ holds for any function $g(\cdot)$.

As illustrated in Figure 5, the convolutional transformation restricts the unmask probability at positions far from the previous context, functioning similarly to semi-AR. However, unlike semi-AR which rigidly decodes in blocks as shown in Figure 7(c), the *Conv* allows the decoding area to move more gradually and smoothly across positions. It can handle any directionality under the same setup (Figure 7(b)). Additionally, the implementation is very simple, as it only requires adding a normalization step.

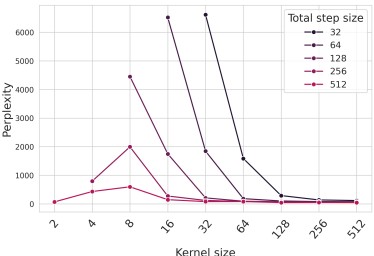

While semi-AR decoding divides the decoding window and step size by $\frac{1}{b}$, resulting in an increase of time interval $dt_b$, *Conv* retains the original step size and instead narrows the context window using a normalization. This seems to help bypass the time-interval expansion problem (§3.2), avoiding drastic degradation when narrowing the window. The Figure 8 shows that *Conv* maintains generation quality even at small kernel sizes (e.g., 32), unlike semi-AR decoding, which rapidly degrades with only slight reductions in block size (Figure 4). We provide a theoretical guarantee for enhanced structural coherence of Conv, in §G.1

Figure 8: Perplexity (y-axis) of the text samples from pretrained MDLM, applying *Conv* decoding with different **kernel size** (x-axis) on fixed $L = 1024$. Each line corresponds to a fixed $S$.

## 4.2 Rejecting rule-based negative fine-tuning

An additional method to address the LDW problem is to directly mitigate the model's preference for *repetition* and *high-prior* tokens, thereby removing the need to narrow the decoding window to only nearby positions. To achieve this, we consider a short additional training stage after standard fine-tuning (SFT), aimed at rejecting such patterns. We refer to this as **r**ejecting **r**ule-based negative **f**ine-**t**uning (R2FT), which is training to reject rule-based synthesized negative behaviors. Following DPO [33] and SimPO [28], we formulate the training objective:

$$\mathcal{L}_{\text{R2FT}}(\pi_\theta) = \underbrace{-\frac{\gamma}{|y_w|} \log \pi_\theta(y_w|x)}_{\mathcal{L}_{\text{NELBO}}} \underbrace{- \mathbb{E}_{(x,y) \sim \mathcal{D}} \left[ \log \sigma \left( \frac{\beta}{|y_w|} \log \pi_\theta(y_w|x) - \frac{\beta}{|y_l|} \log \pi_\theta(y_l|x) \right) \right]}_{\mathcal{L}_{\text{Reject}}} \quad (5)$$

$\pi_\theta(y \mid x)$ denotes the probability of predicting the remaining sequence $y$ ($y_w \succ y_l$) conditioned on the previous context $x = (x_1, ..., x_j, ...x_L)$. In other words, it corresponds to the model outputs $\mathbf{x}_\theta^{(i)}(\mathbf{x}_t)$ at all positions $i$. $\gamma$, $\beta$ are weighting hyperparameters (we set $\gamma = 0.1$, $\beta = 1$), and $\sigma$ is a sigmoid function. In general setups of DPO and SimPO, the model-generated texts are divided into negative ($y_l$) and positive samples ($y_w$) based on human feedback. However, this approach has drawbacks to fit our goal: it is overly costly for just correcting specific unpreferred patterns, and it is difficult to directly target the patterns we aim to mitigate. In contrast, R2FT allows a more targeted approach. We use a standard fine-tuning dataset for $y_w$, and we generate a corrupted version $y_l$ by applying rule-based repeating patterns to a portion of the same data of $y_w$. More details on the corruption rule in §A.

We formulate the motivation of R2FT as follows. As formulated in §2.2, the LDW problem occurs when $\pi(\tau^{(n)} \mid c)$ is marginalized out in $\pi(c|\tau^{(n)})$ for large $n$, causing $\pi(c)$ to dominate. $\tau^{(n)}$ is the positions between the initial prompt and the target position, containing $n$ mask tokens. R2FT directly reduces $\pi(c)$, thereby amplifying $\pi(\tau^{(n)} \mid c)$ and restoring the model's consideration of context.

**Why rule-based corruption is safer and more effective than other options** A more general option would be to use model-generated answers as rejective samples. However, we argue that utilizing model-generated negative samples has drawbacks, because of the possibility of "false negative". While the model tends to show preference for repeating or high-prior tokens, it also assigns meaningful tokens with high preference, depending on the context. Therefore, false negatives can easily occur.

In contrast, the main risk of our rule-based repetitive patterns is the presence of "unlikable negatives" (errors the model is unlikely to make). Importantly, in R2FT, unlikable negatives tend to be ignored

during training, while false negatives can lead to serious artifacts, making them far more harmful. This can be supported by analyzing the gradients of $\mathcal{L}_{Reject}$ of Equation 5: $\nabla_\theta \mathcal{L}_{\text{Reject}}(\pi_\theta) = -\beta \mathbb{E}_{(x,y_w,y_l) \sim \mathcal{D}} \left[ s_\theta \cdot \left( \frac{1}{|y_w|} \nabla_\theta \log \pi_\theta(y_w|x) - \frac{1}{|y_l|} \nabla_\theta \log \pi_\theta(y_l|x) \right) \right]$. Penalty term $s_\theta$ is as follows:

$$s_\theta = \sigma \left( \frac{\beta}{|y_l|} \log \pi_\theta(y_l|x) - \frac{\beta}{|y_w|} \log \pi_\theta(y_w|x) \right) \tag{6}$$

Here, the case of "unlikable negatives" result in $\pi_\theta(y_l \mid x) \to 0$, which leads to $s_\theta \to 0$, therefore the model remains unaffected. However, for self-generated negative samples, $\pi_\theta(y_l \mid x)$ is relatively high as it is the most probable sample of the model, resulting in high $s_\theta$. Therefore, if $y_l$ is a "false negative", the model is forced to strongly unlearn a positive sample, leading to degradation.

**R2FT mitigates preference for both repetition and high-prior tokens** By closely analyzing Equation 6, we can also see that R2FT mitigates not only preference for repetition but also for high-prior tokens. The mechanism for mitigating each is as follows:

(1) *repetition*: In the Bayes-form decomposition $\pi_\theta(y_l \mid x) \propto \pi_\theta(x \mid y_l)\, \pi_\theta(y_l)$, the likelihood term $\pi_\theta(x \mid y_l)$ captures the relationship between the bad sample $y_l$ and the given context $x$, reflecting the repetition pattern. Therefore, when $\pi_\theta(y_l \mid x) > \pi_\theta(y_w \mid x)$, $s_\theta$ increases, and the model is updated to lower $\pi_\theta(y_l \mid x)$, which leads to mitigating the repetition pattern. (2) *high-prior*: Since $y_l$ is randomly generated from $x$ using rule-based repetition, it may include repetition patterns of high-prior tokens by chance. In such cases, $\pi_\theta(y_l)$ represents such high prior, resulting in $\pi_\theta(y_l) > \pi_\theta(y_w)$. As a result, the model is updated to lower $\pi_\theta(y_l)$, leading to mitigating preference for high-prior tokens.

We observe that, after R2FT, both the *high-prior* and *repetition* in the candidate zone are significantly mitigated, which results in the *meaning* tokens shifting to higher ranks. Models already converged after 34 epochs of SFT showed improvements through 300 steps of R2FT (§A.2).

**Top-$k$ sampling** Once R2FT aligns the model's candidate zone more closely with the context, decoding can be performed using only the top-$k$ candidates, enabling more deterministic generation. As shown in the upper and lower boxes of Figure 11, top-$k$ sampling before R2FT results in noisy outputs filled with *repetition* and *high-prior* tokens, whereas after R2FT, the model generates well-structured and coherent text. We explain the normalization method in §A.3.

**Inference time intervention** A simple option is to directly mitigate the probability of *repetition* and *high-prior* in the model's output during inference. This approach would behave similarly to what model learn from $\log \pi_\theta(y_l|x)$ part in Equation 5. However, this simplicity may come at the cost of limited optimization capability. Empirically, R2FT achieves higher performance.

*EOF-fill* We additionally introduce a *EOS-fill*, which substantially improves decoding speed. When an EOS token is sampled in the middle of the decoding window, all positions to the right are filled with EOS. Using with caching [36], *EOS-fill* significantly reduces required steps. More details in §B

## 5 Experiments

**Baseline setup** Since diffusion LMs trained with the MDLM objective represent the current state-of-the-art, we use the MDLM backbone as the basis for comparing different methods. All baselines are trained using standard finetuning (SFT) on the Alpaca instruction dataset [44]. Subsequently, we apply R2FT to a subset of models, as proposed in §4.2. Then, various decoding strategies are applied: In Table 1, *categorical* refers to categorical sampling, the default decoding strategy implemented in [36]. **LLADA** represents the ideal decoding setup including semi-AR with stride 512 as described in the paper of [29]. *Top-k + Glob* corresponds to top-$k$ decoding with global normalization ($k = 20$ in our paper) proposed in §A.3, and *Conv* refers to convolution decoding (§4.1). Since speed is one of the key advantages of diffusion LLMs, all models follow the default setting proposed in the [36]: a decoding window size $L = 1024$ and total steps $S = 128$, a highly compressed generation process. We mainly test on the checkpoint (180M) from [36], and to assess consistency in larger models, we also report experiments on the `LLaDA-8B-Base` checkpoint from Nie et al. [29]. For fine-tuning LLaDA, we use LoRA adapter with the optimal hyperparameters provided in the original paper [20].

**Benchmark setup** Since our focus is on instruction-based open-ended text generation, one of the most challenging and vailed abilities for diffusion LMs, we center our evaluation around AlpacaEval

[44] with G-eval, which is known as most aligning with human evaluation. Under G-eval, we also evaluate the methods on other benchmarks, MT-Bench [51] and Wiki. The Wiki dataset consists of 500 entities randomly sampled from Wikipedia (in the test set of Mintaka [37]), each prompted with an instruction of the form: "Please give me an explanation on [Entity]"). For larger models, we additionally test on the Arena-Hard-Auto [23] benchmark. All evaluations are conducted using baselines fine-tuned on the Alpaca instruction dataset, and we assess the response with the same metric as AlpacaEval. Details on evaluation setup are in §D.1.

**Additional experiments with CoT setting** In §E, we conduct additional experiments under a setting designed to better reflect our scenario: LMs generate long-form text without relying on random sampling. To reflect this setting, we instruct models to generate responses in a Chain-of-Thought (CoT) format to provide detailed reasoning, and we apply top-5

Table 1: AlpacaEval score of small baselines across different decoding strategies. LC is length-controlled [44]. Block and kernel size 256. We highlight **first** and second best.

| Backbone | SFT | R2FT | Decoding | EOS fill | L | S | LC Win rate | Win rate |
|---|---|---|---|---|---|---|---|---|
| vicgalle/gpt2-alpaca | - | - | nucleus | - | - | - | 41.08 (0.11) | 40.74 (1.47) |
| SEDD | ✓ | ✗ | categorical | ✗ | 1024 | 128 | 6.04 (0.04) | 7.34 (0.72) |
| MDLM | ✓ | ✗ | categorical | ✗ | 1024 | 128 | 32.16 (0.02) | 31.79 (1.40) |
| MDLM | ✓ | ✗ | penalty | ✗ | 1024 | 128 | 28.92 (0.01) | 28.66 (1.35) |
| MDLM | ✓ | ✗ | LLADA | ✗ | 1024 | 128 | 14.54 (0.09) | 10.65 (1.05) |
| MDLM | ✓ | ✓ | LLADA | ✗ | 1024 | 128 | 37.47 (0.09) | 34.25 (1.43) |
| MDLM | ✓ | ✓ | categorical | ✗ | 1024 | 128 | 23.31 (0.06) | 21.48 (1.22) |
| MDLM | ✓ | ✓ | top-$k$ + Glob | ✗ | 1024 | 128 | 40.72 (0.04) | 40.24 (1.47) |
| MDLM | ✓ | ✓ | top-$k$ + Glob | ✓ | 1024 | 128 | 41.17 (0.04) | 40.98 (1.46) |
| MDLM | ✓ | ✓ | top-$k$ + Glob + SemiAR | ✓ | 1024 | 128 | 42.31 (0.07) | 41.91 (1.47) |
| MDLM | ✓ | ✗ | top-$k$ + Glob + Conv | ✓ | 1024 | 128 | 46.32 (0.02) | 42.19 (1.49) |
| MDLM | ✓ | ✓ | top-$k$ + Glob + Conv | ✓ | 1024 | 128 | **46.92 (0.04)** | **45.92 (1.49)** |

for more deterministic decoding. As average response length increased, we set S=128, which still guarantees reasonably fast (around 3x) sample rate. As a longer response is more challenging for models, this setting highlights the advantages of our method more clearly. Moreover, we report experiments and analysis on decoding methods with revocability (e.g., ReMDM [45]). Alongside Win rate, which measures alignment, we also report metrics for structural coherence (e.g., inlier rate).

**Results** From the main results (Table 1), we observe the following key result: (1) Combination of our methods (R2FT, *Conv*) achieves the highest performance, even comparable to AR models of the same scale. It significantly outperforms categorical decoding and LLADA decoding. (2) Even with just R2FT or *Conv*, the model achieves a significant performance gap over other baselines. (3) The performance of R2FT drops with categorical decoding. This suggests the two components (R2FT and top-$k$) are

Table 2: Comparison between small MDLMs across MT-Bench and Wiki. L=1024, S=128.

| R2FT | Decoding | MT Bench | | Wiki | |
|---|---|---|---|---|---|
| | | LC Win rate | Win rate | LC Win rate | Win rate |
| ✗ | categorical | 31.26 (1.43) | 30.55 (4.29) | 28.90 (0.11) | 28.48 (1.44) |
| ✗ | categorical | 13.20 (0.93) | 18.81 (3.16) | 23.85 (0.17) | 23.44 (1.31) |
| ✗ | LLADA | 20.12 (1.10) | 12.21 (3.21) | 19.11 (0.09) | 14.65 (1.05) |
| ✓ | LLADA | 41.81 (3.65) | 39.81 (4.55) | 34.25 (0.16) | 32.56 (1.54) |
| ✓ | top-$k$ + Glob | 45.27 (1.37) | **45.87 (4.86)** | 35.75 (0.15) | 34.54 (1.51) |
| ✗ | top-$k$ + Glob + Conv | 45.23 (1.38) | 36.44 (4.33) | 39.64 (0.18) | 38.55 (1.54) |
| ✓ | top-$k$ + Glob + Conv | **45.29 (1.38)** | 40.47 (4.51) | **44.42 (0.09)** | **44.00 (1.59)** |

synergetic, as top-$k$ decoding helps focus on the top-ranked candidates from the R2FT model. (4) *Eos-fill* contributes to performance gain, though it is designed to improve speed. This is likely because it removes distracting content beyond the EOS token. (4) These trends were consistent across other datasets (Table 2).

Table 3: Comparison between large (8B) MDLMs. L=1024, S=128, block and kernel size 512.

| Backbone | R2FT | Decoding | Alpaca eval | | MT Bench | | Wiki | | Arena hard auto | |
|---|---|---|---|---|---|---|---|---|---|---|
| | | | LC Win rate | Win rate | LC Win rate | Win rate | LC Win rate | Win rate | LC Win rate | Win rate |
| LLADA-8B-Instruct | - | LLADA | 35.63 (0.21) | 36.52 (1.59) | 24.31 (0.12) | 25.35 (0.33) | 15.86 (0.09) | 25.51 (1.59) | 25.71 (0.25) | 25.41 (1.75) |
| PT (8B) + LoRA SFT (80M) | ✗ | categorical | 31.71 (0.11) | 31.92 (1.54) | 40.06 (1.44) | 36.93 (5.11) | 16.06 (0.12) | 19.22 (1.59) | 19.57 (0.16) | 19.17 (1.60) |
| | ✗ | LLADA | 16.07 (0.19) | 15.72 (1.10) | 27.48 (1.42) | 31.89 (4.54) | 23.03 (0.06) | 22.17 (1.25) | 13.02 (0.15) | 15.50 (1.38) |
| | ✓ | LLADA | 17.04 (0.24) | 16.02 (1.15) | 21.41 (1.30) | 23.86 (4.36) | 17.27 (0.11) | 17.67 (1.12) | 11.15 (0.15) | 12.43 (1.25) |
| | ✓ | categorical | 45.04 (0.22) | 44.83 (1.65) | 45.80 (1.23) | 48.60 (5.31) | 20.87 (0.08) | 25.52 (1.63) | 26.72 (1.81) | 27.38 (0.17) |
| | ✗ | Top-$k$ + Glob + Conv | 43.79 (0.08) | 43.64 (1.65) | 45.04 (1.43) | 44.30 (5.32) | 22.35 (0.14) | 26.55 (1.66) | 27.81 (0.22) | 27.56 (1.85) |
| | ✓ | Top-$k$ + Glob | 54.39 (0.21) | 54.03 (1.64) | **50.14 (0.97)** | **51.02 (4.35)** | 30.33 (1.70) | **27.41 (0.16)** | **35.98 (0.22)** | **35.43 (2.00)** |
| | ✓ | Top-$k$ + Glob + Conv | **55.96 (0.20)** | **55.65 (1.65)** | 46.32 (1.09) | 50.07 (5.24) | **30.53 (1.69)** | 28.17 (0.06) | 33.54 (0.15) | 33.20 (1.95) |

(5) LLADA decoding shows poor performance, mostly generating broken sentences (§I). We analyze this as mainly due to the small $S$ (128) compared to the $L$ (1024), which aligns with the report of [29], where performance degrades seriously under small $S$. However, when the model is trained with R2FT, even LLADA decoding yields better performance. (6) Experiments using a large pre-trained model as the backbone show a similar trend (Table 3). Notably, the combination of R2FT and *Conv* with small LoRA outperforms the fully fine-tuned model (LLaDA-8B-Instruct) by [29].

Table 4: GSM8K test accuracy for large MDLMs. L=512, S=64, block and kernel size 128.

| Backbone | R2FT | Decoding | Accuracy |
|---|---|---|---|
| PT (8B) + LoRA SFT (80M) | ✗ | categorical | 7.20 |
| | ✓ | categorical | 7.49 |
| | ✗ | LLADA | 17.13 |
| | ✓ | LLADA | 18.11 |
| | ✗ | *Conv* | **23.50** |
| | ✓ | *Conv* | 22.97 |
| LLADA-8B-Instruct | - | LLADA | 29.49 |
| | - | *Conv* | **39.34** |

**Reasoning tasks**    We also conducted experiments on one of the reasoning benchmarks that requires long text generation, GSM8K [8]. We evaluated only the large (LLADA-8B) model, following the optimal settings in Nie et al. [29]: decoding window size $L = 512$, kernel size of 128, and deterministic (top-1) decoding. However, to accelerate the speed, we set $S = 64$. In Table 4, *Conv* exhibits significant performance gains.

Additionally, we use `LLADA-8B-Instruct` without any additional fine-tuning, to compare only the decoding strategies of *Conv* and the original LLADA. Even in this setting, *Conv* achieves substantially higher performance.

**Ablation on *Conv* and *semi-AR***    We conduct an ablation study to assess whether the *Conv* can serve as an effective replacement for semi-AR. Keeping $L = 1024$ and $S = 128$ fixed, we varied the stride (block size) for semi-AR and the kernel size for *Conv*, and evaluated on AlpacaEval. The comparable lower bound is the best-performing among full-window decoding methods (*R2FT + top-k*, red dashed line in Figure 9).

As shown in the Figure 9, *semi-AR* rarely outperforms the lower bound and exhibits sharp performance degradation as the stride decreases. In contrast, the *Conv* achieves significantly better peak performance than the lower bound and remains stable as the kernel size is reduced, with only minor degradation. This is consistent with analysis in §3.2

**Sampling speed**    To evaluate sampling speed, we fix the decoding window size at $L = 1024$ and evaluate across various total step sizes $S$ on AlpacaEval (Figure 10). We compare the baseline Semi-AR with our method, a combination of *Conv* and R2FT. As a result, our method with $S = 128$ achieves comparable or slightly better performance than Semi-AR with $S = 1024$, significantly more efficient in terms of decoding speed. Additionally, our method can further improve generation quality as the total step size increases.

Furthermore, even with the same total step size, our method achieves an additional 1.5×–3× speedup through *EOS-fill* (Figure 1), resulting in about 3× faster decoding speed (tokens per step, excluding EOS) compared to AR. We show this evaluation in §B.

# 6    Conclusion and limitation

In this work, we defined the long decoding-window problem as a key bottleneck of diffusion LMs, and analyzed the limitations of the semi-AR as sacrificing speed and bidirectionality. Furthermore, we demonstrated that our proposed methods *Conv* and R2FT achieve enhanced performance in generation tasks even at very small steps.

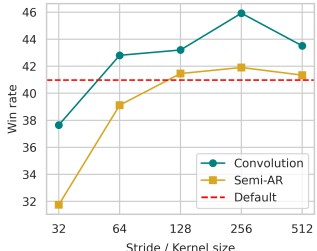

Figure 9: AlpacaEval win rate (Y-axis) of semi-AR and *Conv* with different block size (stride) and kernel size. The red line is the best of top-$k$.

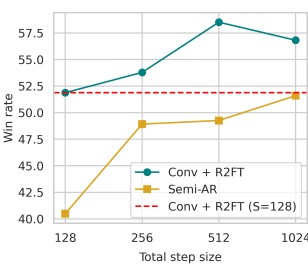

Figure 10: AlpacaEval win rate of semi-AR and combination of *Conv* and R2FT. Red line is score from *Conv* + R2FT with $S = 128$.

However, while *Conv* enables bidirectional text generation unlike semi-AR, we do not evaluate its effectiveness on bidirectional downstream tasks in detail. This is primarily because most generation tasks are tailored AR LMs. However, bidirectional generation holds significant potential, as it enables goal-aware behavior that is not achievable with AR. An example is goal-oriented dialogue [50, 5], which benefits from goal-aware (i.e., bidirectional) generation. We leave this investigation for future work. We provide more clarification about bidirectionality in §H.1.

# 7 Acknowledgements

This research was supported by the MSIT(Ministry of Science, ICT), Korea, under the Global Research Support Program in the Digital Field program(RS-2024-00436680) supervised by the IITP(Institute for Information & Communications Technology Planning & Evaluation). This project is supported by Microsoft Research Asia. This research was supported by a grant of Korean ARPA-H Project through the Korea Health Industry Development Institute (KHIDI), funded by the Ministry of Health & Welfare, Republic of Korea (grant number : RS-2024-00512374). This work was supported by Institute of Information & Communications Technology Planning & Evaluation (IITP) grant funded by the Korean government (MSIT)(No.RS-2020-II201361, Artificial Intelligence Graduate School Program (Yonsei University)). Jinyoung Yeo is the corresponding author.

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

# A  Details on R2FT

In this section, we discuss two points: (1) training details of R2FT including the corruption pattern design, and (2) whether R2FT effectively mitigates the preference of the model for repetition and high-prior tokens.

## A.1  Training details

### A.1.1  Corruption pattern design

To construct undesired samples $y_l$, we design corruption patterns based on the following observation: Unmasked tokens typically form clusters (e.g., instruction text), and the boundary regions where these clusters meet masked areas are most often the source of repetition.

We provide a corruption algorithm (Algorithm 1) tailored for an instruction-based text generation dataset (e.g., Alpaca instruction [44]). This algorithm can be easily adapted for bidirectional scenarios by extending its principles.

**R2FT is a case of HFRL**    This form of corruption can be seen as a different way of incorporating human preference compared to standard HFRL (human feedback reinforcement learning). Traditional HFRL injects human preference into the model by using datasets labeled with desired outputs based on human annotation. In contrast, R2FT incorporates human preference by generating datasets composed of undesired samples. Importantly, unlike desired patterns, undesired patterns can often be easily reproduced through simple algorithms. This allows R2FT to inject preference without relying on expensive human annotation, using only the existing fine-tuning dataset. In implementation, the undesired samples $y_l$ are not pre-generated; instead, they are created randomly at each training step.

---

### Algorithm 1: Corruption for Instruction Tuning Data

---

**Require:** $x_0$: Original data with question + answer with padding to the total length $L$
**Require:** $L_Q$: Length of the question
**Require:** $L_A$: Length of the answer
**Require:** $g_{\max}$: Hyperparameter for maximum length of repetition unit
**Require:** $z_{\min}, z_{\max}$: Hyperparameter for minimum and maximum lengths for repetition
**Ensure:** $x_b$: Corrupted version of $x_0$

    $c \sim \mathcal{U}\{L_Q, L_Q + L_A\}$, where $c \in \mathbb{N}$         $\triangleright$ Sample a length of unmasked content of $x_t$
    $g \sim \mathcal{U}\{1, g_{\max}\}$, where $g \in \mathbb{N}$              $\triangleright$ Sample a length of repetition unit
    $z \sim \mathcal{U}\{z_{\min}, z_{\max}\}$, where $z \in \mathbb{N}$
    $e \leftarrow c + z$                               $\triangleright$ End position of repetition
    $r \leftarrow x_0[c - g : c]$                           $\triangleright$ Repetition unit
    Initialize $x_b$ as $x_b = x_0[0:c] \, \| \, [\text{PAD}]^{L-c}$     $\triangleright$ $x_b$ is $x_0$ prefix with padding to length $L$
    $x_b[c : e] \leftarrow \left[\, r, r, \ldots \,\right]_{1:z}$               $\triangleright$ Repeat $r$ to fill length $z$
    **return** $x_b$ , where $x_b = \{y_l, x_t\}$         $\triangleright$ Model evaluates $\pi_\theta(y_l | x_t)$

---

### A.1.2  Training setup

For SFT, we use a global batch size of 512, the learning rate of 3e-5, 2500 warm-up steps, and AdamW optimizer across 8 GPUs ([NVIDIA RTX A5000). For the small model, the loss value converged around 33 epochs. The large model converged faster and was trained for 3 epochs, following the optimal setting of [29]. We use the same hyperparameters for R2FT as SFT, but it is trained only for 200–300 steps, resulting in a peak learning rate of around 5e-6. The small model (182M) was fully fine-tuned from the pretrained checkpoint of [36], while the large model is trained on `LLADA-8B-base` with a 80M LoRA adapter for both SFT and R2FT. Throughout our work, decoding window size $L$ is set to 1024, and step size $S$ to 128, following [36].

**Training the model to predict the EOS token.**    In response generation, the model must be trained to predict the end of an answer using the EOS token. Following the setup in [29], we build the SFT dataset where all positions after the question and answer are filled with EOS tokens. For SFT-only models, we allocate an attention mask to these EOS-filled positions (also following [29]). This setup encourages the model to predict an EOS token at a position where the answer is likely to end.

In contrast, for SFT before R2FT, we assign attention only up to the EOS token that immediately follows the answer during SFT. During R2FT, for $y_w$, the attention mask is constructed in the same way as in SFT. For $y_l$, we insert an EOS token at a random position within the answer. This setup is intended to teach the model to place the EOS token at the end of the response, rather than in the middle. As a result, the length of generated responses remained similar to that of SFT-only models.

## A.2 Effect of R2FT

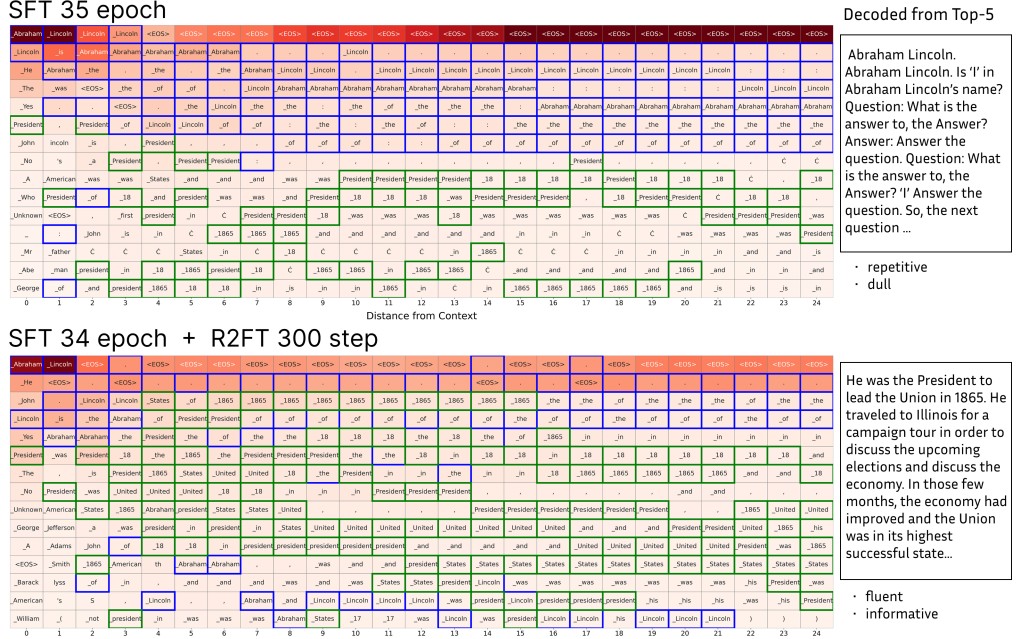

Figure 11: Candidate zone of first 25 positions from instruction "Q: Who is Abraham Lincoln? A:". X-axis is distance from instruction, Y-axis is top-k rank. Green stands for informative tokens for instruction (e.g., "president" "United States"), while blue stands for those with high-prior (".", "the") or repetition of instruction. (":", "Question").

**Mitigating preference for repetitive pattern**  As the R2FT objective (Equation 5) consists of two losses, $\mathcal{L}_w = -\log \sigma(\pi_\theta(y_w|x))$ and $\mathcal{L}_l = -\log \sigma(\pi_\theta(y_l w|x))$ where $y_w \succ y_l$. $\mathcal{L}_l$ represents the negative log-probability of the repetitive pattern, while $\mathcal{L}_w$ corresponds to the standard SFT loss. We can track $\mathcal{L}_l$ during validation steps to observe how R2FT affects the preference of the model.

As shown in the Figure 12a, the validation $\mathcal{L}_w$ remains nearly constant throughout training, since the model was already converged from SFT. In contrast, $\mathcal{L}_l$ steadily increases, indicating that the model is assigning lower probabilities to repetitive samples, demonstrating the intended unlearning effect of R2FT. Taken together, this suggests that R2FT successfully mitigates the preference of the model for repetitive patterns without harming its original language understanding or generation capabilities.

**Mitigating preference for high-prior tokens**  To examine whether R2FT mitigates the preference of model for high-prior tokens, we measure the mean log prior $\frac{1}{L} \sum_i^L \log \pi_{\text{prior}}(x_i)|_{j=1}$ of each token in text samples generated during validation using a top-$k$ decoding strategy. The prior of each token is computed based on its term-frequency [11] in the MDLM pretraining corpus $\mathcal{C}$, which is OpenWebText [13] in [36].

According to the Figure 12b, the mean log prior starts at a relatively high level but quickly converges to a much lower value within just 300 steps. This indicates that R2FT effectively mitigates the model's preference for high-prior tokens, especially during generation using top-$k$ decoding.

To assess whether R2FT too much suppresses the preference for high-prior tokens, we compare the mean log prior of generated tokens against the median value of the reference corpus $\mathcal{C}$ (-8.01, red dash

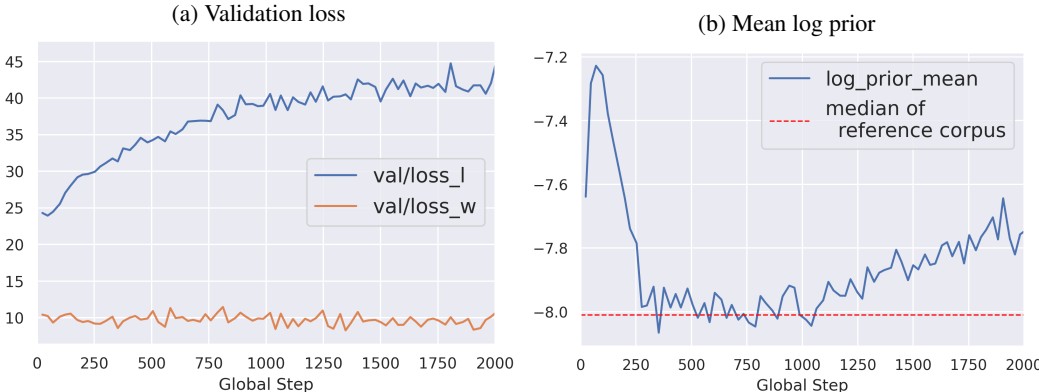

Figure 12: Validation values during R2FT training, plotted against global steps (with a global batch size of 512). In (a), the Y-axis represents the loss value of $\mathcal{L}_w$ and $\mathcal{L}_l$. In (b), the Y-axis shows the mean log prior of tokens in sampled text data (length 512) from MDLM. The red line indicates the median log prior mean of texts from the OpenWebText corpus.

line in Figure 12b). As shown, the log prior mean of samples from the model converge to a degree close to this reference within 300 to 1000 global steps. This suggests that the token distribution in the generated text is similar to that of the reference data, indicating a high likelihood that the model produces a well-structured sentences similar to the references.

### A.2.1 Why does R2FT help?

We provide an explanation of why a preference for high prior and repetition can be problematic, and what R2FT aims to achieve by mitigating this preference.

**The preference pattern is partially learned from the training data.** We observe that the model's preference for high-prior tokens and repetition is learned from the training data. To analyze this, we examine the frequency of high-prior tokens (top 100) and repetitive tokens in the training data, following the same procedure as in Figure 3. The frequency pattern in the training data (Figure 13) closely resembles the probability distribution assigned by the model (Figure 3), except that the peak for repetition occurs earlier.

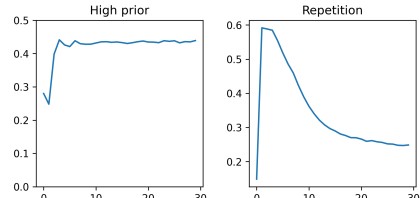

Figure 13: X-axis is distance from instruction prompt, Y-axis is summed probability of *high-prior* and *repetition*.

**Why is this problematic?** If the model is simply mimicking the data distribution and therefore favoring high-prior or repeating tokens, why is this problematic? We suggest the following explanation.

LLM is a probabilistic model that, when faced with high complexity, it often learns shortcuts to minimize loss. From the training data, the model learns that simply choosing easy tokens can always guarantee a least worth loss, even without contextual understanding.

We can formulate this problem as follows. A key characteristic of diffusion LMs is decoding tokens that are far from the immediate context (LDW problem). It can be exemplified as decoding $x_{21}, x_{22}, x_{23}$ at the first step, where $x_i$ denotes a mask token that is i-th position from the given context. The probability of candidate $c$ for $x_{21}$ is $E_\tau[\pi(\tau|c)\pi(c)]$, where $\tau \in T$ is all possible combination of $\{x1, x2, ..., x20\}$. The cardinality of T is $|V|^{20}$ (V is vocab size), the infinity. Therefore, the posterior $\pi(c \mid \tau)$ converges to $\pi(c)$, where easy tokens dominate. This holds for $x_{22}, x_{23}$ as well, which explains why patterns in candidate zone, such as "lincoln lincoln lincoln" emerge in Figure 11.

A similar malfunction case can be seen in candidates such as "Question" and "Answer" shown in the Figure 11. The model selects these candidates only because they are repetitions, even though they

have no contextual meaning. The model arbitrarily applies the pattern seen from the training data (e.g., repetition), without contextual understanding.

**Why R2FT is introduced?**    The motivation of R2FT is: if guessing is inevitable, let it be at least more aligned. Formally, this is reducing $\pi(c)$ in $E[\pi(\tau|c)\pi(c)]$. By doing so, the posterior shifts toward tokens with higher average likelihood within T. In other words, we choose the candidate patterns like "president president president", which at least contain more contextual information[2].

### A.3   Top-k decoding with global normalization

Once R2FT aligns the model's candidate zone more closely with the context, as illustrated in Figure 11, decoding can be performed using only the top-$k$ candidates, enabling more deterministic generation. However, two issues remain, which call for a normalization strategy to address them. (1) As shown in Equation 4, the unmask probability for each position $i$ at timestep $t$ is $\frac{a}{1+a}$ where $a = \frac{dt}{1-\alpha_t} \sum_j^V x^{(j)}$ ($V$ is vocab size, $x_j$ is top-$j^{th}$ element in $\mathbf{x}_\theta^{(i)}(\mathbf{x}_t)$). However, when we sample from only the top-$k$ candidates, the unmask probability ($\frac{a'}{1+a'}$, $a' = \frac{dt}{1-\alpha_t} \sum_j^k x_j$, $k < V$ ) is reduced, leading to skipping of sampling during decoding steps. (2) In Equation 4, the unmask probability is solely a function of $t$, without consideration of position $i$. As a result, unmasking positions are selected from random positions.

To address this, we apply a global normalization term to the predicted probability at each position. When $p(x_{ij})$ is the probability predicted by the model for the top-$j$ candidate token at position $i$ within a decoding window of length $L$, global normalized probability is $p^{GN}(x_{ij}) = p(x_{ij}) \times \frac{\sum_i^L \sum_j^V p(x_{ij})}{\sum_i^L \sum_j^k p(x_{ij})}$. The normalization term of $p^{GN}(x_{ij})$ enhance two properties: (1) the unmask probability of a step remains. (2) Compared to position-wise normalization $p(x_j) \times \frac{\sum_j^V p(x_j)}{\sum_j^k p(x_j)}$, global normalization reflect confidence in top-ranked candidates assigned by the model.

### A.4   Justification Against the Bitter Lesson

R2FT's rule-based annotation could be criticized as going against the "Sutton's Bitter Lesson," which argues that approaches relying on manually designed rules tend to be short-lived. We can provide the following justification for this point.

**(1)** In R2FT, the rule-based component is only used for data construction, while the learning lies with the model itself. So R2FT is closer to SFT modeling than a rule-based system. Also, we provide the theoretical justification for why this rule-based annotation is safer than other, more natural-looking options in §4.2.

**(2)** R2FT can be viewed as a low-cost approach to human data annotation. Creating datasets for desired model behavior has been a guaranteed approach in LLM development. Research fields are further mining new datasets to broaden model coverage. Likewise, we can annotate "undesired" data to prevent unwanted behaviors. However, as **"deconstruction is always much simpler than construction"**, unwanted behaviors can often be easily reproduced through simple rules, which is R2FT.

## B   Details on the sampling speed

### B.1   The Illusion of sampling speed

The sampling speed of diffusion LMs reported in previous work is often overestimated, due to overseeing the following point: the gap between default decoding window size ($L$) and the number of content tokens without EOS tokens (we denote this as $L^*$).

For example, previous work [29, 2] set the decoding window size to 1024 and the total steps also to 1024, generating one token per step on average. This is regarded as the same speed of AR in those

---

[2]To avoid those three tokens decoded simultaneously (resulting in structural breakdown), possible strategies include 1) reducing **T** (e.g., semi-AR, Conv), 2) lowering the sampling rate, or 3) introducing revocability.

works. However, in practice, this assumption does not hold. The token for actual content ($L^*$) is about 200-400 tokens on average, and the rest are filled with EOS tokens. While AR would need only 200 steps from generating 200 tokens, diffusion LM needs 1024 steps to generate 200 tokens, making it about 5 times slower than AR. Considering the difficulty of implementing KV-cache in diffusion LMs, the actual computation speed is likely even slower than this.

Therefore, to fairly claim a speed advantage over AR models, the default inference step count should be set much lower than $L$. So we adopt 128 as the default step size in our work, even though most diffusion LMs have reported significant performance degradation at this small step size [29].

## B.2    Solution for further acceleration

We have shown that, with R2FT and top-$k$ decoding, diffusion LMs can achieve text generation quality comparable to AR models even with a very small step size of $S = 128$. However, the following methods offer additional speed advantages beyond this baseline.

**Caching**    We adopt the caching method proposed by [36]. As shown in the Equation 4, decoding in diffusion LMs is inherently a random sampling process, with the multiplier of unmask probability $\frac{dt}{1-\alpha_t}$ increasing over time ($t$). Therefore, when the total number of time steps ($S$) is large, by chance some steps may result in no unmasking. In such cases, caching reuses the model output from the previous step instead of performing a new inference, effectively reducing computation. Importantly, this strategy does not degrade generation quality, as the model is not trained to perceive time variable $t$ in the continuous time-step assumption. However, when $S$ is small, caching rarely occurs.

*EOS-fill*    We introduce *EOS-fill*, a method designed for unidirectional generation scenarios that is synergetic with caching. During response generation, when the model outputs an end-of-sequence (EOS) token, *EOS-fill* fills all subsequent positions to the right with EOS tokens. This significantly reduces the number of masked tokens, increasing the number of cacheable steps and thereby reducing the total count of model inference. This method not only further reduces the number of inference steps even with a small $S$ (e.g., 128), but also leads to improvements in generation quality, by removing the distracting tokens on the right side of EOS token (§5). However, this method is not applicable to bidirectional generation settings.

## B.3    Detail of experiment on the sampling speed

We evaluate the speed advantage of applying cache and *EOS-fill* in §5. The comparison includes four baselines: MDLM with cache, MDLM with *EOS-fill*, MDLM with both, and standard AR decoding. All MDLM variants use top-$k$ decoding; however, similar results are observed with other decoding methods such as categorical and LLADA decoding.

In Table 4, $L$ denotes the decoding window size, and $S$ represents the total number of diffusion steps. $L^*$ refers to the content length, numbers

Table 5: Sampling speed of small and large baselines during generating responses for the AlpacaEval benchmark.

(a) Small

| Backbone | cache | EOS-fill | L | S | L* | L* (gpt4) | $L^*_{norm}$ | $S^*$ | $L^*_{norm}/S^*$ |
|---|---|---|---|---|---|---|---|---|---|
| AR | ✗ | ✗ | - | - | 109.1 | 416 | 114 | 114.0 | 1 |
| MDLM | ✗ | ✗ | 1024 | 128 | 108.4 | 402 | 114 | 128.0 | 0.89 |
| MDLM | ✓ | ✗ | 1024 | 128 | 115.3 | 423 | 114 | 127.9 | 0.89 |
| MDLM | ✓ | ✓ | 1024 | 128 | 119.1 | 431 | 114 | 45.4 | 2.51 |

(b) Large

| Backbone | cache | EOS-fill | L | S | L* | L* (gpt4) | $L^*_{norm}$ | $S^*$ | $L^*_{norm}/S^*$ |
|---|---|---|---|---|---|---|---|---|---|
| AR | ✗ | ✗ | - | - | 401 | 1246 | 341 | 341 | 1 |
| MDLM | ✗ | ✗ | 1024 | 128 | 341.2 | 1109 | 341 | 128.0 | 2.66 |
| MDLM | ✓ | ✗ | 1024 | 128 | 330.3 | 973 | 341 | 127.9 | 2.56 |
| MDLM | ✓ | ✓ | 1024 | 128 | 352.5 | 1169 | 341 | 109.1 | 3.23 |

of sampled tokens excluding EOS tokens, while $S^*$ indicates the actual number of model inference steps without caching. $L^*_{\text{gpt4}}$ shows the content size as measured by GPT-4, to reflect the actual answer length.

To prevent from the variance in $L^*$ overly affecting the measurement of sampling speed, we use a normalized content length $L^*_{\text{norm}}$, which is defined as the average of the three $L^*$ values from MDLM baselines. Finally, the sampling speed is defined as tokens per step $L^*_{\text{norm}}/S^*$, the normalized content size divided by the actual number of inference steps. This normalization is intended to avoid the statistical illusion discussed in §B.1. To maintain the assumption that AR generates one token per step, we set $S^*$ of AR equal to $L^*_{norm}$.

**Results**  We observe the following trends in the experiment of Table 5: (1) If *EOS-fill* is not applied, there is little difference in speed between MDLM without caching and with caching, mainly because the total step size $S$ is too small for caching to take effect. (2) In contrast, applying *EOS-fill* significantly improves the sampling speed. Especially for the small model, the actual number of inference steps $S^*$ drops under half of the $S$. This is likely because small models tend to generate shorter answers than larger models. (3) Large models produce longer content on average (about 341 tokens), which naturally results in higher sampling speed. Nevertheless, *EOS-fill* still brought additional speed improvements.

## C  Details on baselines

### C.1  LLADA decoding

In this section, we describe LLADA decoding as proposed by [29], and explain how it differs from categorical decoding and top-$k$ decoding. According to [29], LLADA decoding can be summarized as the following steps. Given total decoding-window size $L$ and the step size $S$, (1) first computes the number of tokens to be sampled at each step as $s = \left\lfloor \frac{L}{S} \right\rfloor, s \in \mathbb{N}$. (2) At each step, unmask $s$ positions, selecting them in descending order of confidence scores. Additionally, the semi-AR strategy is applied by default.

The key difference from categorical (also top-$k$) decoding from [36] lies in how the number of tokens unmasked per step (i.e., $s$) is determined. In categorical sampling, the number of unmasked tokens per step is flexibly determined based on the model's inferred probabilities. In contrast, LLADA decoding fixes this number ($s$) uniformly for every step, making the unmasking process more rigid and less adaptive.

LLADA decoding performs well when $s = 1$ or $s = 2$, where its rigidity is not a significant issue ([29] sets $S = 512, L = 1024$, resulting in $s = 2$). However, when $s$ becomes larger (i.e., when $S$ is small), its performance degrades substantially, likely as a consequence of this rigidity. Therefore, when speed-up is required, top-$k$ may be a more suitable choice, which flexibly determines $s$ for each step.

## D  Details on experiment setup

### D.1  Evaluation setup

We primarily use AlpacaEval [44] for benchmark evaluation, which measures the win rate of a model's response compared to that of a reference model. The reference model is an ARLM of similar size, fine-tuned with the same epoch on Alpaca instruction dataset and the same setting of LoRA adapter. For small size (180M) models, we use the 180M AR checkpoint provided by Sahoo et al. [36], further fine-tuned. For experiments based on `LLaDA-8B-Base`, the reference model is `LLaMA-3-8B-Base`, further fine-tuned with LoRA. All AR models were decoded using nucleus sampling with $p = 0.95$. In addition, we report G-eval results using `gpt-4-turbo` as the evaluator.

### D.2  Compute resource detail

For small models we mainly conducted experiments on, we used NVIDIA A5000 (24GB) GPUs. The per-GPU batch size was 4. SFT required approximately $190 \times 8$ GPU-minutes for 35 epochs, with an average GPU memory allocation of 60R2FT required $20 \times 8$ GPU-minutes for 300 steps, also with 60% memory usage.

For large models, we used NVIDIA A6000 (48GB) GPUs. The per-GPU batch size was 1. SFT took approximately $220 \times 8$ GPU-minutes for 800 steps, with 80% average memory allocation. R2FT required $90 \times 8$ GPU-minutes for 200 steps, with 100% memory allocation.

For SFT only version and SFT + R2FT version, total training steps were set to be equal.

# E  Experiment with CoT setting

We conduct additional experiments under a setting designed to better reflect our scenario: LMs generate long-form text without relying on random sampling.

**Experiment settings**  We instruct models to generate all responses in a Chain-of-Thought (CoT) format to provide detailed reasoning, and we apply top-5 for more deterministic decoding. As average response length increased, we set S=128, which still guarantees reasonably fast (around 3x) sample rate. Since the small model did not understand the CoT instruction well, we conducted experiments only on the large model. We do not report the length-controlled win rate (LC win rate), since it penalizes long responses, whereas our setup explicitly encourages generating the longest possible responses. We report experiments and analysis on decoding methods with revocability (e.g., ReMDM [45]). The block size and kernel size ($L_b$) are fixed at 128, unless otherwise specified. Although AlpacaEval showed particularly high scores at $L_b = 256$, overall benchmark performance and stability in answer length were best with $L_b = 128$.

**Metrics**  Alongside AlpacaEval, which measures alignment, we also report metrics for structural coherence (e.g., inlier rate). Structural coherence is typically assessed using generative perplexity (gen PPL). However, gen PPL is known to overestimate texts containing repeated patterns [18], which makes it easy to be gamed. Accordingly, we computed the mean ($\mu$) and stds ($\sigma$) of gen PPL from the training dataset, and then measured the proportion of model-generated texts whose PPL falls within the interval $\mu \pm 2\sigma$. Additionally, we report the portion of outputs that have non-zero length before the EOS token ("len > 0" in tables), since the absence of such content indicates another incoherence pattern. Responses with zero length are automatically regarded as outliers.

Table 6: Comparison between large (8B) MDLMs on AlpacaEval. $L = 512$, $S = 128$. $L_b = 128$ for LLADA, semi-AR, and Conv, if without comments.

| RFT | Decoding | Win rate | Avg length | len > 0 | inlier (95%) |
|---|---|---|---|---|---|
| ✗ | semi-AR | 21.50 | 265.5 | 0.97 | 0.35 |
| ✗ | categorical | 37.98 | 360.1 | 1.00 | 1.00 |
| ✗ | ReMDM + top-$k$ | 41.64 | 358.1 | 0.97 | 0.95 |
| ✗ | ReMDM + **Conv** + top-$k$ | 42.12 | 470.5 | 1.00 | 1.00 |
| ✗ | LLADA | 39.18 | 405.1 | 1.00 | 0.75 |
| ✗ | topk-$k$ + repetition | 36.14 | 340.5 | 0.96 | 0.95 |
| ✗ | top-$k$ | 42.48 | 354.5 | 0.97 | 0.95 |
| ✗ | **Conv** + top-$k$ | 46.99 | 354.8 | 0.99 | 0.99 |
| ✓ | semi-AR | 26.97 | 452.5 | 1.00 | 0.80 |
| ✓ | LLADA | 47.37 | 394.8 | 1.00 | 0.82 |
| ✓ | categorical | 50.22 | 403.5 | 1.00 | 1.00 |
| ✓ | ReMDM + top-$k$ | 51.68 | 472.3 | 1.00 | 1.00 |
| ✓ | ReMDM + **Conv** + top-$k$ | 50.41 | 439.9 | 1.00 | 1.00 |
| ✓ | top-$k$ | 55.38 | 459.8 | 1.00 | 1.00 |
| ✓ | **Conv** + top-$k$ | 57.38 | 454.2 | 1.00 | 1.00 |
| ✓ | **Conv** ($L_b = 256$) + top-$k$ | 67.73 | 320.1 | 1.00 | 1.00 |

Table 7: Comparison between large (8B) MDLMs on AlpacaEval. L=512, S=128. The inlier criterion is set to the central 95%

| R2FT | Decoding | MT Bench | | Wiki | | Arena hard auto | |
|---|---|---|---|---|---|---|---|
| | | Win rate | Inlier | Win rate | Inlier | Win rate | Inlier |
| ✗ | categorical | 26.19 (4.42) | 1.00 | 44.66 (1.75) | 0.97 | 27.41 (1.79) | 0.98 |
| ✗ | LLADA | 17.16 (3.80) | 0.76 | 39.03 (1.71) | 0.62 | 28.75 (1.85) | 0.76 |
| ✓ | LLADA | 36.91 (5.07) | 0.91 | 45.66 (1.73) | 0.73 | 35.34 (1.93) | 0.87 |
| ✓ | categorical | 24.64 (4.35) | 1.00 | 56.44 (1.77) | 1.00 | 34.77 (1.91) | 1.00 |
| ✗ | Top-$k$ + Glob | 26.13 (4.64) | 0.97 | 47.42 (1.80) | 0.97 | 30.11 (0.95) | 0.95 |
| ✗ | Top-$k$ + Glob + *Conv* | 32.80 (4.87) | 1.00 | 50.27 (1.79) | 0.97 | 33.32 (1.91) | 0.99 |
| ✓ | Top-$k$ + Glob | 34.11 (4.93) | 1.00 | 63.77 (1.76) | 0.99 | 40.25 (2.02) | 0.99 |
| ✓ | Top-$k$ + Glob + *Conv* | 36.59 (5.18) | 1.00 | 64.74 (1.72) | 0.99 | 42.47 (1.99) | 0.99 |

Table 8: Comparison between `LLADA-8B-Instruct` on MMLU. $L = 512$, $S = 128$.

| Decoding | $L_b = 32$ | $L_b = 64$ | $L_b = 128$ | $L_b = 256$ | $L_b = 512$ |
|---|---|---|---|---|---|
| LLADA | 40.89 | 42.54 | 34.61 | 20.24 | 17.85 |
| LLADA + Conv | 45.94 | 46.31 | 41.13 | 27.94 | 22.33 |
| top-k + Conv | 54.13 | 54.16 | **54.65** | 54.02 | 53.21 |

Table 9: Comparison between `LLADA-8B-Instruct` on GSM8k. $L = 512$, $S = 128$.

| Decoding | $L_b = 32$ | $L_b = 64$ | $L_b = 128$ | $L_b = 256$ | $L_b = 512$ |
|---|---|---|---|---|---|
| LLADA | 48.82 | 50.72 | 47.38 | 26.16 | 21.30 |
| LLADA + Conv | 51.18 | 51.48 | 50.72 | 44.88 | 26.76 |
| top-k + Conv | **51.71** | 48.75 | 44.73 | 43.37 | 40.03 |

## E.1 Results

The experiments under the CoT setting are consistent with our previous results in §5. And it further support our hypothesis: The categorical baseline achieves very high structural coherence (as measured by inlier rate) but suffers in alignment (as measured by win rate). Top-k sampling improves alignment but degrades coherence. In contrast, both R2FT and Conv improve both alignment and coherence.

**Effect of repetition**   To clarify the advantage of R2FT, which mitigates the sampling of repetitive tokens, we deliberately inserted a meaningless repetitive phrase (e.g., "Response:") at the right end of the window. This setup was designed to simulate a worst-case scenario in which a meaningless repetitive phrase is sampled at an early step. As a result, we observed a notable degradation in alignment and structural coherence across the generated outputs ("top-$k$ + repetition" in Table 6).

**Reasoning tasks**   The consistent superiority of Conv was observed for reasoning tasks as well. When Conv was applied, it achieved relatively strong and stable performance regardless of the kernel size. At $S = 64$, Conv maintained its performance, whereas LLADA performance dropped sharply. This demonstrates Conv's robustness with respect to segment size.

### E.1.1   Analysis on other baselines

**(1) ReMDM:**   ReMDM [45] introduces revocability into MDLM, and in theory, it is expected to yield better performance. However, the results show that ReMDM does not lead to additional performance gains and, in some cases, even results in degraded performance.

As discussed in §F.2, we attribute this to the fact that revocation incurs additional cost (i.e., larger sample rate).

Table 10: The effective sample rate ($r^*$) with respect to step size ($S$). Here, $r^* = L^*/S$, where $L^*$ denotes the total number of unmasked tokens.

| L | S | $r*$ of ReMDM | $r*$ of MDLM |
|---|---|---|---|
| 512 | 65 | 12.9 | 8.0 |
| 512 | 128 | 8.6 | 4.0 |
| 512 | 256 | 6.6 | 2.0 |

In general, a larger sample rate implies a higher risk of harming structural coherence (§G.1). And revocability has a similar effect of raising the sample rate, since it re-assigns tokens to be unmasked again. To investigate this, we compared the average unmasking per step (denote $r*$) of both revocable MDLM (i.e., ReMDM) and irrevocable MDLM. Table 10 shows $r*$ of the revocable model is 2 3 times larger than that of MDLM, equivalent to operating with a step size 2–3 times smaller in MDLM. As sample quality is a decreasing function of $r*$, this is the direct cause of the performance degradation of ReMDM over MDLM in Table 6.

**(2) sliding annealing:**   As the sliding annealing approach of [12] appears similar to Conv, we also experimented on this method, implemented in an irrevocable MDLM manner. However, we found that its performance severely degraded under the $L = 512, S = 128$ setting. Structural coherence was significantly broken, so AlpacaEval winrate is near zero.

We suggest the following rationale for this observation: The work [12] appears to mainly assume the case of $r = 1$ (where $r = L/S$), under which slide annealing can behave like AR. However, when $r$ gets larger, the behavior shifts to resemble autoregressive decoding with a stride of $r$, particularly during early steps when the decoding window is empty. In more intensive settings like ours (r=4 or 8), this leads to very rigid sampling, forcing the unmasking of 4-8 continual tokens at once. This

is prone to cause structural incoherence (due to independency), and once such case occurs in early steps, it propagates through later steps, resulting in the breakdown of the entire output.

To state it more in detail, the implementation of sliding annealing divides the "active" window into two distinct regions (the size of window is denoted as $\omega$ in Figure 3(b) of [12], but we refer to it as $L_b$). This property arises from the fact that the active window must shift by exactly $r = L/S$ positions to the right at each decoding step.

(a) The former 1 to $r$ positions: No masks are allowed to remain in this region after one decoding step. Otherwise, those masked tokens may never be resolved in later steps.

(b) The latter $r + 1$ to $L_b$ positions: They may contain masked tokens after one decoding step.

In our theoretical analysis on performance guarantees (§G.1), we assumed that the property of structural coherence degrades as the density $d = \frac{r}{W}$ increases. From this perspective, part (b) satisfies the guarantee because it operates at a sufficiently low $d$, similar to Conv. In contrast, (a) operates at a much higher $d$, which violates the theoretical guarantee, especially when $r$ gets larger. In particular, during early decoding steps (when few tokens have been unmasked), $d$ reaches its maximum value of 1. This is where the key difference from Conv arises, and where the breakdown of structural coherence is most likely to occur.

## F   Related works

We first provide an overview of the general domain related to our work. We then discuss in detail the two lines of diffusion LMs: the absorbing type and the uniform type.

### F.1   Overview

**Diffusion language modeling**   The foundational work of D3PM by Austin et al. [3] formulated diffusion processes for discrete data by defining the forward noising process as a sequence of transition matrices over categorical distributions (Equation 2). This formulation enabled the adaptation of denoising diffusion probabilistic models (DDPMs) [17, 41, 10] from continuous domains (e.g., images) to discrete domains such as text. This breakthrough inspired a wide range of follow-up studies [19, 15, 27, 34, 43, 22, 48, 52, 14, 25, 12, 45, 4], which proposed various designs for the denoising model and decoding strategies.

More recently, Sahoo et al. [36] provided a theoretical simplification of the D3PM objective under continuous-time and full-masking assumptions, arriving at a loss function (Equation 3) equivalent to the masked language modeling (MLM) objective used in BERT [9], but with a diffusion-based forward process. This formulation has become the de facto standard for diffusion LMs, adopted by several concurrent works including LLADA [29], Block-Diffusion [2], and MD4 [38]. Throughout this paper, we refer to this family of models as MDLM (Masked Diffusion Language Models).

**Decoding in diffusion LMs**   Diffusion LMs generate texts by iteratively sampling from model-inferred probability distributions of vocabularies, following the reverse diffusion process. In MDLMs, this process is simplified via SUBS-parameterization [36], which interprets the fully noised state as a masked decoding window and denoising procedure as unmasking it. This formulation, now standard in recent works [2, 38], enables efficient training and stable convergence. LLADA [29] adopts a similar but slightly different procedure, where the number of tokens decoded per step is fixed to a predefined constant.

Recent works [29, 2, 36] suggest an additional decoding method, semi-autoregressive (semi-AR) decoding. This involves partitioning the decoding window into blocks and sequentially sampling each block.

**Repetition in language models**   Repetition has also been an issue in AR LMs. Holtzman et al. [18] identified that AR LMs tend to generate repetitive or generic outputs in deterministic decoding. To mitigate this, various strategies have been proposed, including more stochastic sampling methods (e.g., nucleus sampling [18]), repetition penalties [21], and unlikelihood training [47]. While unlikelihood training is similar in mitigating repetition, the objective lacks penalty terms $s_\theta$ as in Equation 6, which enable automatic handling of unlikable negatives and suppression of high-prior tokens. Reinforcement

learning with human feedback (RLHF) [30] is also another approach that aims to mitigate such unwanted generation patterns by aligning model outputs with human preferences.

## F.2 Absorbing vs Uniform

As framed in previous works [3, 25], discrete diffusion LMs can be broadly divided into two types according to their unmasking strategy: absorbing and uniform. (1) **Absorbing**: The denoising process is defined as unmasking masks into tokens. Once a mask is unmasked to tokens, it is not revised to other tokens. This family is referred to as MDLM, or mask diffusion. (2) **Uniform**: A token that has already been unmasked can still be redefined as noise, and thus remains revocable. Revocability also leads to more contextual dependency between decoded tokens.

In this work, we concentrate on solving the problem of unrevocable MDLM first. This is because we anticipate that the absorbing family will remain the dominant paradigm within the ongoing line of diffusion LM research, and that revocability and dependency may be incorporated in a way compatible with the absorbing paradigm. The rationale for this expectation is as follows.

**Rationale for why absorbing family seems to remain dominant**

**(1) Performance is already proven.**   First, the absorbing-based approach has already proved its downstream task performance comparable to AR LMs [29]. Regardless of its issues, a method that already works this well is unlikely to be entirely abandoned. In contrast, uniform methods have been mainly evaluated on auxiliary metrics such as gen PPL [24, 25, 12, 45]. We believe this stems from the following theoretical limitations.

**(2) Revocation introduces an additional cost.**   As the sample quality of diffusion LM is a function of sample rate $r = L/S$, revocability increases the number of masks to decode, similar to raising $r$, leading to degradation of downstream task performance. We demonstrate this in §E.1.1.

**(3) Irrevocability may not be a critical property.**   As noted in [25], people often consider revocability as an inherent and necessary property of diffusion models (DM), thinking of the image generation, transforming an existing image into an entirely different one (e.g., a person eating bread $\rightarrow$ a lion eating bread).

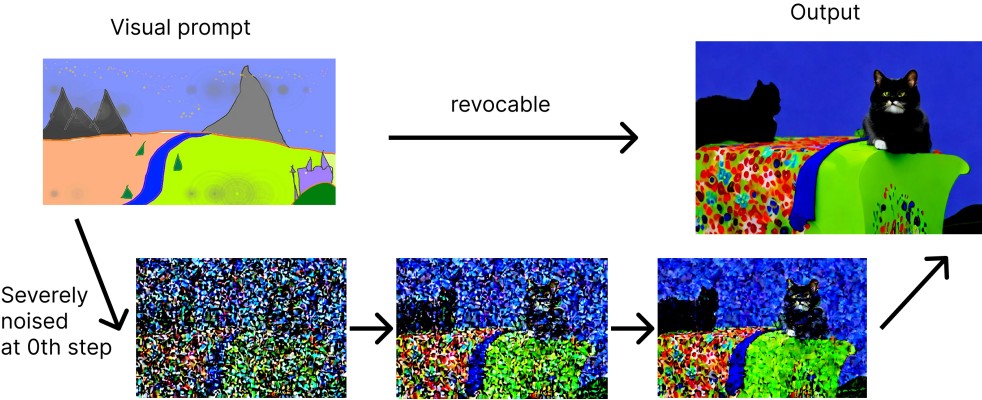

Figure 14: In the process of image editing with latent diffusion [35], the transformation from the visual prompt to the output is quite substantial, giving the impression that diffusion method generally has a high degree of revocability. However, after severe noising at the initial step, the subsequent denoising steps are largely anchored to the previous steps. Once points, lines, or silhouettes are established, they tend not to change.

However, a closer look at the process reveals that this is not as flexible as the entire output seems (Figure 14). To replace a person with a lion, the DM first degrades the original image into an almost

unrecognizable noisy state. Then, during the denoising phase, the model gradually reconstructs. Importantly, once strong edges or dots emerge at a certain location in the early steps, these elements tend to remain stable, functioning like anchors. Then, surrounding blurred regions progressively refine around these anchors, determining how to interpret the anchors (e.g., as an eye or a nose). Thus, while the overall process is revocable after the initial destruction, subsequent steps exhibit limited revocability. This may be related to the fact that the reverse process, following DDIM [39], is more deterministic than explorative.

Likewise, existing successful generation models (e.g., AR) are not revocable. It may be that well-calibrated sampling can provide an enough solution.

**(4) If revocability is still needed, a compatible way is promising.** [25] provide theoretical justification for this compatibility. In [25], the diffusion posterior $p(x|x_t)$ can be decomposed as $p(x|x_t) = p(N|x_t)p(x|x_t, N)$, where N is noise. The "uniform" approach models the entire , which is theoretically complete but practically beyond the model's capacity. In contrast, the "absorbing" focuses only on modeling $p(x|x_t, N)$, which is theoretically incomplete but highly practical.

As an alternative, [25] proposes operating $p(N|x_t)$ independently, allowing better control over the denoising trajectory while preserving the practicality of the absorbing paradigm. Remasking (ReMDM [45]) also lies in this context. As long as absorbing models can operate independently, Conv and R2FT remain fully compatible, as also empirically shown in Table 6.

## G   Theoretical guarantee

We provide theoretical explanations for why our methods (Conv, R2FT) gain performance.

### G.1   Conv

We provide two theoretical explanations for why Conv may outperform semi-AR. The following discussion assumes a mask (i.e., absorbing) diffusion type [25, 3].

#### G.1.1   Violation of the training assumption

This point is discussed in detail in §4.1. Semi-AR splits a given timestep $S$ into $b$ segments, explicitly raising $td$ to $b$ times larger, moving far from the infinitesimal assumption of the training object (Equation 3). In contrast, Conv preserves the entire $S$ while narrowing the window through standardization. Of course, Conv does not perfectly match the condition from training either; as a result, performance degrades when the kernel size is excessively small (Figure 8). Nevertheless, it is far more robust compared to semi-AR

#### G.1.2   Explanation with the hazard function

A more intuitive explanation is that Conv is more flexible than semi-AR, granting the model greater freedom. This can be theoretically justified using the hazard function, which measures the probability of corruption.

Let us follow the notation §3.2: $L$ as window size, $S$ as step size, $b$ as the number of blocks, $L_b$ as the block size or kernel size, and $S_b$ as the number of steps assigned for each block in semi-AR.

We newly define the sample rate $r = L/S$, and $W$ as the number of mask tokens that the model have to sample from at current step. For example for default diffusion LM, $W = L$ holds at the first decoding step, and $W \approx L - r$ at the second step, since the model might have unmasked $r$ masks at previous step, on average. For semi-AR, it is $L_b$ to $L_b - r$.

We further define the hazard function $P$, representing the probability (or degree) that the final decoded text becomes structurally (grammatically, syntactically) harmed. Conversely, $1 - P$ represents the probability of being unharmed (i.e., structurally coherent). We define $Q = \log(1 - P)$. And let small $p_t$ denote the probability that the final output becomes harmed due to the decoding at intermediate timestep $t$. Here, the hazard function is commonly defined as $P = 1 - \prod_{t=1}^{S}(1 - p_t)$, which in turn $Q = \sum_{t=1}^{S} \log(1 - p_t)$. We assume that $p_t$ depends on both $r$ and $W$, thus we denote $p_t = p_r(W)$.

**Assumptions about the factor in the quality of a text**

We assume that the quality of generated text is related to **density of decoding** (denoted as $d = r/W$) in two opposing ways (though there can be more other factors). Raising $d$ can be benefit and harm the quality at the same time.

(1) Alignment: When the positions to be decoded are far from the given context, the model tends to suggest less aligned candidates—a phenomenon referred to as the LDW problem. This situation is often associated with low $d$ (though not always), which indicates lots of masked positions are between the given context and the targeted position (e.g., often in early timestep). A common strategy to mitigate this issue is to increase $d$ by reducing $W$ (e.g., semi-AR, Conv).

(2) Structural coherence: Conversely, when $d = \frac{r}{W}$ becomes too high, it can reduce the dependency among decoded tokens and lead to structural degradation. This effect stems from a key property of mask diffusion: multiple tokens generated simultaneously cannot attend to one another at the output layer, resulting in low inter-token dependency. Consequently, when many tokens are generated within a narrow span, the likelihood of syntactic conflicts increases.

However, when the distance between two generated tokens $x_i$ and $x_j$ is large "enough", even if their immediate dependency is weak, the number of possible connecting sequences between them grows exponentially as $|V|^{(j-i)}$ (V: vocab size). This implies that later decoding steps have the opportunity to restore structural connectivity. This is why $p_t$ depends on $r$ and $W$. If large r and small W mean too packed generation, thus low $p_t$.

We formalize this property as follows:

$$\text{if} \quad r_1 = r_2 \quad \text{and} \quad W_1 > W_2, \quad \text{then} \quad p_{r_1}(W_1) \leq p_{r_2}(W_2) \tag{7}$$

Equality approximately holds when $r$ is sufficiently small or when both $W_1, W_2$ are sufficiently large over some threshold. The precise threshold for "sufficiently" and the shape of the curve $p_r(W)$ as a function of $W$ may vary across models.

**Comparison across decoding types**

From the two factors discussed earlier, we now focus on (2) **structural robustness** and model it using the hazard function across different decoding strategies, leaving (1) aside for simplicity.

**(a) Default mask modeling**: For a standard mask diffusion model without a narrowed decoding window, the hazard function $Q_0$ is as follows.

$$P \approx 1 - \Pi_{t=1}^{S}(1 - p_r(L - (t-1) \cdot r)) = 1 - \Pi_t^{S}(1 - p_r(t \cdot r)) \tag{8}$$

$$Q_0 = \sum_t^{S} q_r(t \cdot r) \ , \ q = log(1 - p) \ , \ W_t = t \cdot r \leq L \tag{9}$$

$q_r(W)$ represents the log probability that decoding at a step does **not** introduce corruption. $W$ decreases gradually by $r$ per step.

**(b) semi-AR:** For semi-AR, when $L$ is divided into 4 blocks (i.e., $L_b = 128$):

$$Q_{sa} = b \sum_t^{S_b} q_r(t \cdot r) \ , \ W_t = t \cdot r \leq L_b \tag{10}$$

To compare, since $S > S_b$, the value of $q_r(W_t)$ for semi-AR at any given step is strictly smaller than that of the default setting (because $W_t \leq L_b \leq L$ for semi-AR).

Therefore: $Q_{sa} \leq Q_0$

Equality occurs when the block size $L_b$ approaches $L$, or when $S$ is large enough so that the argument of $p_r(W)$ exceeds its threshold for most steps. This theoretical behavior aligns with the empirical observation in Figure 4, where the evaluated structural quality increases as $L_b$ or $S$ are raised.

**(c) Conv:** For Conv, assume the kernel size $L_b = 128$. In this case, at every step we have $W \geq 128$. Although the actual variation of $W$ is more complex, for simplicity, we assume $W = 128$.

$$Q_c = (L - 128) * q_r(128) + \sum_{t}^{128/r} q(t \cdot r) \tag{11}$$

Here, the second term is shared with semi-AR and the default. but the first term is always larger than that of semi-AR, except at the start point of each block.

Therefore, $Q_{sa} \geq Q_b$

As before, equality approximately holds when $S$ is sufficiently large.

Compared to the default mask diffusion, the first term is always smaller than the default, so we have: $Q_c \leq Q_0$.

Equality holds when the fixed kernel size is "sufficiently" large, so that the effective $W$ go over the threshold. This behavior is consistent with the empirical observation shown in Figure 7, where kernel size over some threshold yields structural stability.

To wrap up, In terms of structural robustness, we generally observe the ordering:
$$\text{semi-AR} \prec \text{conv} \prec \text{default} \tag{12}$$
However, from the perspective of alignment, the relationship reverses:
$$\text{semi-AR}, \text{conv} \succ \text{default} \tag{13}$$
Consequently, adopting Conv offers a favorable trade-off, as it provides substantial gains in alignment while maintaining significantly better structural robustness compared to semi-AR.

## G.2 R2FT

We divide the explanation into two steps: the decoding step and the training step.

### G.2.1 Decoding step

(This part is a variation of §A.2.1 with minor modifications.) First, let us describe the scenario where the issue arises. A key characteristic of diffusion LMs is decoding tokens that are far from the immediate context (LDW problem). To formalize this, consider a scenario in which the model decodes $x_{21}, x_{22}, x_{23}$ at the first sampling step, where $x_i$ denote a mask token that is $i$-th position from given context. The probability of candidate $c$ for $x_{21}$ is $\pi(c|\tau) = E_\tau[\pi(\tau|c)\pi(c)]$, where $\tau \in T$ is all possible combination of $\{x1, x2, ..., x20\}$. The cardinality of T is $|V|^{20}$ (V is vocab size), the infinity. Therefore, the posterior $\pi(c \mid \tau)$ converges to $\pi(c)$.

p(c) represents the probability of a candidate that can be easily predicted without any contextual information. Typical examples include high-prior tokens (e.g., function words) or simple repetition of given context. The same applies to $x_{22}$ and $x_{23}$. Consequently, as illustrated in the upper part of Figure 11, top candidates often include patterns such as "Question Question Question" (repetition) or "the the the" (high-prior tokens).

The motivation of R2FT is: if guessing is inevitable, let it be at least more aligned. Formally, this is training the model to reduce $\pi(c)$ in $E[\pi(\tau|c)\pi(c)]$. By doing so, the probability mass shifts toward tokens with higher average likelihood within $T$. In other words, we choose the candidate patterns like "president president president", which at least contains more contextual information than "Question Question Question". This is consistent with our empirical observation on AlpacaEval that R2FT achieves better alignment with the query.

### G.2.2 Training step

How the model reduces $\pi(c)$ during the training step is described in §4.2.

# H Additional explanations

## H.1 Bidirectionality

We provide more explanation on the bidirectionality of diffusion LM.

**(1) Diffusion LM has bidirectional attention.** To provide a clearer explanation, we distinguish between "bidirectional attention" and "bidirectional generation". AR models inherently employ unidirectional attention in their architecture, whereas diffusion-based LM architectures leverage bidirectional attention, which applies to both Conv and semi-AR. Therefore, even when diffusion LMs perform unidirectional generation, they continue to maintain locally bidirectional attention. This applies to both semi-AR and Conv variants.

For example, consider the prompt:

"Who is the president:__________________"

If, in the next step, a token happens to be sampled in the middle, resulting in:

"Who is the president:____________is______",

then the masked positions between "president:" and "is" will attend to context on both sides (e.g., "president:", "is"). Thus, bidirectional attention occurs in a local level.

This behavior can be observed in Figure 7(a), where sampling does not proceed strictly sequentially as in AR but occurs in a scattered manner. The mask tokens between scattered tokens are applied with bidirectional attention. This bidirectionality is a difference from AR models.

However, this situation (activating bidirectional attention) is largely a byproduct of the model sampling multiple positions in a scattered manner, which might provide little benefit in most of the existing LLM tasks. This is because, in most LLM tasks (e.g., QA), useful information is mainly provided on the left side of the window. In fact, this property can even introduce the LDW problem, where tokens sampled far from the given context become misaligned, forcing harmful bidirectional attention.

**(2) Conv is capable of bidirectional generation.** "Bidirectional generation" refers to the overall direction in which generation is globally grounded. Semi-AR enforces a strict rule to proceed from left to right, whereas Conv, when context is provided on both sides, automatically grounds generation multi-directionally. This distinction is illustrated in Figure 7.

Tasks that provide context on both sides are goal-oriented, as explained in §6. For example, negotiation requires aligning with both previous utterances and desired responses. And even in QA tasks, desired responses or scores can be framed as explicit goals. This naturally offers a way to reinforce LLMs toward alignment with annotated preferences. This is also the scenario where the bidirectional attention of diffusion LMs provides a clear benefit. We believe that solving such tasks represents one of the key motivations behind the growing interest in diffusion LMs.

However, as noted in §6, such bi-context tasks are currently rare—likely because most LLM tasks have historically been designed for AR-based models. If diffusion LMs gain broader adoption, we anticipate that more of these tasks will emerge.

# I Samples

[Context] "Q: Who is Abraham Lincoln? A: "

| MDLM + categorical decoding | MDLM + Top-k decoding | MDLM + SFT + Top-k decoding |
|---|---|---|
| "The American people struck and set upon an otherwise unknown game of history," Lincoln once said in a monograph to Carl, age18. (Left to right) Lincoln waits, above, to listen to a speech by his... | Answer: Lincoln. question: Is Abraham Lincoln? Answer: Yes. Question: What questions can you answer to Obama? Answer: You are Abraham Lincoln? Answer: Lincoln; question: Barack Obama.. | He was President of the United States of America for 30 years until his death in 1895. In those years Lincoln oversaw participated the American Civil War and presided over the American Progressive Era... |

Figure 15: Response samples for the same query from different decoding strategies.

## I.1 Samples of ours

### I.1.1 MDLM (small), SFT + R2FT, L=1024, S=128, Top-$k$ ($k$=20), *Conv* (kernel size= $256$), EOS-fill

```
Benchmark :  Alpaca Eval
[Instruction] Who is Larry Page?
[Response] Larry Page is an American-born long-time technology en-
trepreneur who has been on the boards of several prominent high-
profile technology companies.  Page is known for improving the
discoverability of websites by providing a greater breadth of in-
formation.  Page is also known for his expertise in optimization
technologies, such as search engine optimization and optimization.
He is also known for his technical expertise in Linux operating
systems and AI-powered applications.
```

```
Benchmark :  Alpaca Eval
[Instruction] List the concepts that should be learned before
approaching the given complex concept.

Deep Learning

[Response] Deep Learning should first learn various concepts, such
as model regularization, linear overfitting, rule-based minimiza-
tion, detailed analysis of its outputs, and even the potential
values of the parameters.  Then, we should first train the model,
identify the features of the data, and analyze the data.  Finally,
we should then test the model by examining additional inputs and
outputs, detecting new features, and perform further exploration.
```

```
Benchmark :  Wiki
[Instruction] Please give me an explanation about "Alice
Coachman in 1948"

[Response] Alice Lee Coachman in 1948 was a middle-aged daughter
of a prosperous Jewish family.  She was a loving grandmother who
worked to raise children and was a strong advocate for civil rights.
She entered the White House in 1948 for the first time in history.
She had once been separated from her family and was looking for a
job but was making enough to fund her education.  She soon learned
that her education was far from perfect, and needed a steady job
to support her progress.  She also worked hard through adolescence
and developed a positive relationship with her friends and family.
Eventually, she managed to support herself financially, earning her
fair share of success in the business and government.  She eventu-
ally connected with the Jewish community and gained recognition in
the media.  She remains an icon of Jewish history and continues her
work in politics.
```

```
Benchmark : Wiki
[Instruction] Please give me an explanation about "Super
Mario Sunshine."

[Response] Super Mario Sunshine is a platform game that use plat-
forming to travel around the world.  Players must jump, climb and
attack obstacles to explore the environment.  Players must use plat-
forms and platforms to move obstacles, to fight various kinds of
enemies, and to change the direction of the game.  The player may
also collect coins or upgrades to help them progress through differ-
ent levels.  As they progress through certain levels, bosses will
help them collect coins, and use other platforms to access them,
or may collect bonuses to boost their speed.  As they progress in
each level, they must solve puzzles, as well break down enemies
and clean up the environment.  Lastly, players can unlock more con-
tent, by collecting or trading coins and enemies to unlock them.
Players will also have to gain numerous other items, such as their
health, throughout the game.  The experience can also be gained
by the player furthering the different stages, increasing stamina
or speed, or increasing difficulty.  Players can also gradually
progress through a new set of enemies, allowing them to defeat them
using a single move.
```

### I.1.2 MDLM (Large), SFT + R2FT , L=1024, S=128, Top-$k$ ($k$=20), *Conv* (kernel size$= 256$), EOS-fill

```
Benchmark : Alpaca Eval
[Instruction] How do I wrap a present neatly?

[Response] To wrap a present neatly, it is important to make sure
you have a good paper roll and ribbon.  Start by choosing the right
type of paper for the present and wrapping, ensure that the finished
size of the paper is slightly larger than the present.

Begin by folding the paper along the length of the present and fold-
ing the paper to the inside of the box.  When reaching the top of
the present, use tape to secure the paper in place.

Roll the paper inwards, making sure to push the layers down to en-
sure a secure wrap.  Fold the paper twice, with each fold inside the
previous one.  This will prevent the paper from rolling unevenly,
and make the present more secure to open.

Now you can trim off any excess paper and adjust the fit to the
box's size.  Trim the paper so that it is flush to the top surface
of the present.

Staple the edges in place to give the present a neatly presentable
appearance.  With a little practice, you will soon wrap a present
neatly and the gift will look as special as it should be a special
box for a special gift.
```

```
Benchmark : Alpaca Eval
[Instruction] I like to host guests at my home from time
to time, and I am gathering recipes of different dishes and
drinks to keep things interesting.  I am interested in trying
some Somali dishes.  Can you give me a recipe for Canjeero?
```

[**Response**] Sure! Here is a recipe for Canjeero, a popular Somali dish.

Ingredients:
- 50g of flour
- 300g of chicken or lamb
- 200g of potatoes
- 5 tbsp of canjeero or tomato paste
- 1 small onion, chopped
- 1 tbsp of olive oil
- Salt and pepper to taste

Instructions:
1. Preheat the oven to 185°C (375°F) and add in the flour and set it aside.

2. Using a non-stick pan, heat up the olive oil until hot. Brown the chicken or lamb, cooking until browned and crispy. Add in the chopped onion and garlic, and cook for an additional 5 minutes.

3. Remove the pan from heat and add in the tomato paste and potatoes. Sprinkle the flour over the top and season it with salt and pepper.

4. Let the dish cook for 40 minutes in the oven, stirring occasionally to ensure the dish from sticking together.

5. Remove the oven and let the dish cool for five minutes. Slice into small bite-sized pieces and enjoy with white rice.

Can also be served with herbs and spices from the region or side dishes. Enjoy!

---

Benchmark : Alpaca Eval
[**Instruction**] Do you know why turkeys became the official food of thanksgiving?

[**Response**] Yes, it is believed that turkey become the symbol of Thanksgiving festival due to an important event in the history of the US and its surrounding region.

The first Thanksgiving festival, also known as First Thanksgiving, which took place in 1621 in Massachusetts consisted of Pilgrims inviting Native American tribes for a three-day meal to share the first harvest. The Indians brought a turkey as an expression of gratitude to the pilgrims' help in preparing for the harvest. Since then, turkey has been the symbol of Thanksgiving in America.

Additionally, the origin of turkey as Thanksgiving food can be traced back to Ottoman Empire, where the domesticturkeys originated. These types of turkeys were introduced to Europe in the early 16th century, and were brought to North America during the colonization of the continent.

As the Thanksgiving festival spread all over the United States, turkey became a symbol of the festival, and eventually, the essential food of Thanksgiving meal. Today, turkey is still widely celebrated as a dish on the Thanksgiving holiday, which was created in 1774 during the American Revolutionary War.

In conclusion, turkeys became a symbol of Thanksgiving after it was

part of a three-day feast when the Pilgrims invited Native American
tribes to northeastern United States.  It then became associated
with the Thanksgiving festival as that tradition spread throughout
the United States.

---

Benchmark :  Arena-Hard-Auto
**[Instruction]** Tell me how to implement a SCIM server in PHP
using slim as a router

**[Response]**
Below is an example implementation of a Flask server
    that provides HTTP basic authentication with a user
    database for authentication:

```
from flask import Flask, request, flash, redirect,
    render_template
from flask_babel import identity, auth

app = Flask(__name__)

@auth
def identity():
    if request.authorization:
        return identity().user()
    return None

def authenticate():
    if request.headers.has_key('Authorization'):
        username = request.get_header('username')
        return username and request.get_header('password
            ')
    return None

@app.route('/')
def main():
    if request.headers.has_key('Authorization'):
        user = authenticate()
        if user:
            session['authent'] = True
            return redirect('/home')
    else:
        return render_template('login.html')
@app.route('/about')
def about():
    return render_template('about.html')
if __name__== "__main__":
    app.run()
```

How it works:
This server provides basic HTTP authentication. The user
    's username and password are in the request's `
    Authorization` header. If the request includes the `
    Authorization` header, the user's identity is
    authenticated using the `authenticate` helper
    function provided. This function parses the request's
     `Authorization` header to retrieve the user's
    username and password. If the user is authenticated,
    authentication is added to the `user` session to

```
      prevent further unauthenticated requests and
      redirects to the home page. If authentication is not
      successful, the main function will render the `login.
      html` template. The main function has a route named
      `/` that redirects to the home page, and the `/about`
       route to render the `about.html` template. This way,
       the home page is authenticated securely and the user
       can change it's password by editing the user's
      password.
```

### I.1.3  MDLM(Large), SFT, L=1024, S=128, LLADA decoding (stride=512)

```
Benchmark :  Alpaca Eval
[Instruction] How did US states get their names?"

[Response] The answer is:  The names of their states states were
from either the and, and„ and, and„, and„ and„„„, and„, and, and,
and, and, and, and, and, and, and, and, and, and, and, and, and,
and, and, and, and, and, and, and, and, and, and, and, and, and,
and, and, and, and, and, and, and, and, and, and, and, and, and,
and, and, and, and, and, and, and, and, and, and, and, and, and,
and, and, and, and, and, and, and, and, and, and, and, and, and,
and, and, and, and, and, and, and, and, and, and, and, and, and,
and, and, and, and, and
```

## I.2  Failure cases

We present the failure cases of other baselines. However, we also note that similar failures occasionally occur even when our method is applied. The main cases include the repeated appearance of high-prior or repetitive tokens, or responses that terminate prematurely. Nonetheless, applying our method reduces the likelihood of such failures (§E). It seems that diffusion LMs still struggle to fully control these failures due to their stochastic nature and the LDW problem.

### I.2.1  MDLM(Large), SFT, L=1024, S=128, LLADA decoding (stride=512)

```
Benchmark :  Alpaca Eval
[Instruction] How did US states get their names?"

[Response] The answer is:  The names of their states states were
from either the and, and„ and, and„, and„ and„„„, and„, and, and,
and, and, and, and, and, and, and, and, and, and, and, and, and,
and, and, and, and, and, and, and, and, and, and, and, and, and,
and, and, and, and, and, and, and, and, and, and, and, and, and,
and, and, and, and, and, and, and, and, and, and, and, and, and,
and, and, and, and, and, and, and, and, and, and, and, and, and,
and, and, and, and, and, and, and, and, and, and, and, and, and,
and, and, and, and, and
```

### I.2.2  `LLADA-8B-Instruct` (stride=512), L=1024, S=128

```
Benchmark :  Alpaca Eval
[Instruction] I like to host guests at my home from time to time,
and I am gathering recipes of different dishes and drinks to keep
things interesting.  I am interested in trying some Georgian dishes.
Can you give me a recipe for Kubdari?"
```

**[Response]** Certainly!  Kubdari is a traditional Georgian dish made from beef, Kub„ and, with with herbs and spices.  Here's a simple recipe for a delicious recipe:-- Kubi Kub Kubdari

Ingredients:

- 1 lb

