# OpenReview forum: "Fast and Fluent Diffusion Language Models via Convolutional Decoding and Rejective Fine-tuning"
_NeurIPS.cc/2025/Conference — NeurIPS 2025 spotlight_

### Official Review · Reviewer_TLks · 2025-06-18

**Clarity:** 2
**Significance:** 3
**Originality:** 3
**Rating:** 5
**Confidence:** 4

**Summary:**

This work seeks to improve mask-based diffusion language models in the "highly compressed generation" regime: each model call generates many tokens on average (e.g., 8). The work identifies failure modes in SOTA models when they're pushed to these extremes, conjecture on the causes/mechanisms underlying these failure modes, and propose concrete solutions in the form of Rejecting Rule-based negative Fine-Tuning (`R2FT`) and Convolutional decoding (`Conv`). `R2FT` is a reinforcement-learning-inspired method to penalize the model for generating samples that do not satisfy some hard rules. `Conv` is like block diffusion, but smoother. Experiments show that, together, these two mechanism improve on the baselines in this "highly compressed" regime.

**Questions:**

### (Not a question.) Clarification on "*independently* and *irrevocably*"

Suppose that the next two tokens are either "admirable aadvark" or "benevolent beaver", with equal probability. However, "admirable beaver" and "benevolent aadvark" are **impossible**. When $S$ is large, an MDLM is likely to generate those two tokens on distinct steps: the first token can be sampled 50/50, and the second token will later be deterministic. If the two tokens are to be predicted on the same step (which is likely when $S$ is small), then there is no way for the MDLM to guarantee a valid generation: AB and BA will occur as often as AA and BB. In view of this situation, [1] breaks the independence assumption; [2,3,4,5] can fix their mistakes.


### Question 1: Do you agree that the conjectured causes/mechanisms in Section 2.2 are incomplete, premature and/or overstated?

*(This is my main issue with the manuscript.)*

Line 113:
> In Figure 2, we observe excessive repetition of tokens from the previous context (e.g., “Question”, “Answer”, “:”), which is not helpful in the current step. This phenomenon likely stems from the inherent tendency of LLMs to repeat previous context

The training set likely contains Q&A data. In those datasets, a question is followed by an answer, *then by another question and another answer etc.* The MDLM is trained to predict those structures at a global level, not just to answer questions. The model may not so much "repeat previous context" as "reproduce a pattern seen in training".

Lines 124:
> The key property is that, as the distance from the previous context gets farther, the meaning zone tends to fall below the repetition and high-prior zones. This trend is also consistent with statistical observation (Figure 3). We measure the sum of probabilities of high-prior (top-100) and repetition (tokens overlapping the prompt) tokens at varying distances from the instruction prompt.

My prediction is that the same "zones" would be observed in the training data, particularly with Q&A datasets (i.e., get the statistics for the tokens that follow question contexts in the raw dataset, as you would find it in a github repo).

(Line 131)
> However, they peak after around 5-10 positions.

What is the typical length of an answer in the training set? My bet is that it is "around 5-10 positions". If this were to be confirmed, it would further support my point about the model just doing its job of learning the training set. In this view, part of what `R2FT` does is to train the model to deviate from that training distribution, to satisfy our specific practical needs.

Taking a step back, like in the "(admirable|benevolent) (aardvard|beaver)" case, it is fine that "Lincoln" has a high independent probability to come up in consecutive positions: the problem is that we independently sample multiple tokens at once, which may cause "Lincoln Lincoln Lincoln Lincoln" to be generated. To be clear, I'm not saying that my own perspective is "better": like the authors, I'm making conjectures of my own. **My point is that there are alternative explanations** compatible with the presented evidence: the proposed causes/mechanisms are incomplete, premature and/or overstated. I see two possible paths forward:

1.1: **Tone down** these conjectures, presenting alternative interpretations and noting that some of these observations may be specific to Q&A (or similar) tasks.

1.2: **Double down** by gathering more data, disproving the alternative interpretations that I bring up above, and testing the "zones" hypothesis for arbitrary contexts (or better qualify what kind of context this applies to).

Do you agree with my viewpoint? Would you consent to revise an eventual camera-ready accordingly?


### Question 2: Can you comment or rebut my point concerning the specificity of `R2FT` to Q&A-like tasks?

I said that, in its current format, this fix appears rather specialized to Q&A and similar tasks. I acknowledge that, *a priori*, `R2FT` could be generalized to expanded rulesets, but having humans leverage their domain knowledge to craft such rulesets would be against the "Bitter Lesson" that general methods leveraging computation have historically shown to be ultimately the most effective by a large margin.

What are your thoughts? You may also suggest how more general rulesets may be obtained automatically, and/or provide evidence that such more general rulesets are not needed.


### Question 3: Can you say a little more about what `R2FT` "does" to the probability distribution?

In my view, instead of dropping the independence and/or irrevocability assumptions, `R2FT` alters the independent token distributions to improve the coherence of a given generation algorithm. Do you agree with this take? If yes, what form does this "alteration" take? From Figure 11, I can see that the temperature has increased for latter tokens, and the blue "high prior" options have lowered in rank. But this is only a snapshot before the inference process has started: how different would Figure 11 be after the token "Lincoln" has been selected at distance 1 from context? Have you measured the $D\_{KL}$ between these two models at different steps during generation? I welcome any data/observations in this general direction of inquiry.

### Question 4: Can you clarify when `Conv` refers to the uni/bi-directional version?

My current understanding/guess is that, except in Figure 7, every instance of `Conv` refers to the bidirectional version. Is this the case? In any case, this should be crystal clear in an eventual final version.

**Ethical Concerns:**

["NO or VERY MINOR ethics concerns only"]

**Final Justification:**

The authors have agreed to make changes to the wording of their observations, as well as clarifying the mechanism through which their RL operates. Some misunderstandings remain (see https://openreview.net/forum?id=HvIRFV0J90&noteId=4v6V1jYx1E ), but no further iteration is required on my end.

I support the publication of this work.

**Limitations:**

yes

**Paper Formatting Concerns:**

.

**Quality:**

3

**Strengths And Weaknesses:**

- The issues identified in Section 2.2 are real and important. I have myself experienced such failures modes, and I am not aware of existing work reporting them so explicitly. I believe that it is good for this information to be made more broadly known.
- Section 2.2 alternates between *stating observations* about failure modes and *conjecturing causes/mechanisms*, and I judge these latter conjectures to be incomplete, premature and/or overstated.
- In my personal opinion, the fact that multiple tokens are sampled *independently* and *irrevocably* is an important cause of the failure modes observed by the authors in that regime. I'm aware of at least one work [1] that attempts to fix the issue by lifting the independence assumption, and many recent papers [2,3,4,5] conversely lift the irrevocability assumption. I believe that this situation should be properly discussed.
- My understanding is that `R2FT` lifts neither assumptions of independence and irrevocability, but instead uses a novel third way: alter the independent token distributions to improve the coherence of a given generation algorithm.
- In its current format, this fix appears rather specialized to Q&A and similar tasks. *A priori*, `R2FT` could be generalized to expanded rulesets, though this would go against Rich Sutton's Bitter Lesson, so I would not bet on it too much on the long term. Nonetheless, realizing the existence of this novel third approach is what made me revisit the significance score from "fair" to "good".
- Although the unidirectional version of `Conv` (Figure 7(a)) appears to be very similar to the "slide annealing" from [3], the bidirectional version (Figure 7(b)) appears novel to me.
- The evaluations are mainly pitted against variants of MDLM. I believe that these results suffice to answer "yes" to the question "can `R2FT+Conv` improve on a given MDLM?", but does not give a clear picture of how the resulting model scores in the fast-growing ecosystem of diffusion models.

[1] Liu et al. Discrete Copula Diffusion. https://arxiv.org/pdf/2410.01949
[2] Liu et al. Think while You Generate: Discrete Diffusion with Planned Denoising. https://arxiv.org/pdf/2410.06264
[3] Fathi et al. Unifying Autoregressive and Diffusion-Based Sequence Generation. https://arxiv.org/abs/2504.06416
[4] Wang et al. Remasking Discrete Diffusion Models with Inference-Time Scaling. https://arxiv.org/abs/2503.00307
[5] Boget et al. Critical Iterative Denoising: A Discrete Generative Model Applied to Graphs. https://arxiv.org/abs/2503.21592

---

> ### Author Rebuttal · Authors · 2025-07-31
>
> We are deeply impressed by the reviewer’s profound knowledge and insightful comments. Addressing these questions was an enjoyable experience. All discussions will be incorporated into the camera-ready version.
>
> ---
>
> **[Q-1] *Why is the preference for high priors and repetition problematic, and what does R2FT aim to achieve? (Regarding your Question1, 3)***
>
> = First, let me briefly respond to the statistical questions:
>
> $ Average length of response in training data:
>
> 57.47 (GPT-2 tokenizer)
>
> $ Does a single instance in the training data consist of multiple question–answer pairs?
>
> No. The behavior (“Question” and “Answer” appearing on top candidates) is not directly learned.
>
> $ Do the probability patterns for repeating and high-prior tokens in the training data resemble those in Figure 3?
>
> In the training dataset, the probability of high-prior and repetitive tokens (we will note them “**easy tokens**”) resembles the pattern of the model, declining after an initial peak. However, the peak occurs much earlier (within 1~2 positions). This suggests that the model’s distribution behaves slightly differently, likely due to more influence of conditional likelihood at the positions near the context. We will contain the figure in the camera-ready.
>
> This observation might lead you to three questions.
>
> **[Q-1-1]  If the model is simply mimicking the data distribution and therefore favoring high-prior or repeating tokens, why is this problematic?**
>
> =  LLM is a probabilistic model that, when faced with high complexity, it often learns shortcuts to minimize loss. From the training data, the model learns that simply choosing easy tokens can always guarantee a least worth loss, even without contextual understanding.
>
> We can formulate this problem as follows. A key characteristic of diffusion LMs is decoding tokens that are far from the immediate context (LDW problem). It can be exemplified as decoding $x_{21}, x_{22}, x_{23}$ at the first step,  where $x_i$ denote a mask token that is i-th position from given context (similar to the “Lincoln” example). The probability of candidate $c$ for $x_{21}$ is $E_\tau [ p(\tau|c)p(c)]$ , where $\tau \in T$  is all possible combination of $\{x1,x2,...,x20\}$. The cardinality of T is $|V|^{20}$ (V is vocab size), the infinity. Therefore, the posterior $p(c \mid \tau)$ converges to $p(c)$, where easy tokens dominate. This holds for $x_{22}, x_{23}$ as well,  which explains why patterns in candidate zone, such as “lincoln lincoln lincoln” emerge.
>
> Such a problem is not a matter of “independently decoding” property. This is because, regardless of how much a model accounts for dependency with surrounding context on the output layer, no model can feasibly consider $|V|^{20}$ possibilities in advance.
>
> **[Q-1-2] On the other side, why do tokens like *"Question"* and *"Answer"*, which were not even present in the training dataset, appear in the candidate zone?**
>
> = This is a good example of the stated problem in Q-1-1. The model chooses these candidates only because it is repetition, though it has no contextual meaning.
>
> **[Q-1-3] Then, what does R2FT do?**
>
> The motivation of R2FT is: if guessing is inevitable, let it be at least more aligned. Formally, this is reducing $p(c)$ in  $E[p(\tau|c)p(c)]$. By doing so, the posterior shifts toward tokens with higher average likelihood within T. In other words, we choose the candidate patterns like "president president president", which at least contains more contextual information. To avoid those three tokens decoded simultaneously (resulting in structural breakdown), possible strategies include 1) reducing T (e.g., semi-AR, Conv), 2) lowering the sampling rate, or 3) introducing revocability as you mentioned.
>
> (We will show p(c) in each step for both SFT and R2FT in camera-ready, regarding your Question 3)
>
> **[Q-1-4] Isn’t it rather a problem of irrevocability and independence? Isn’t Conv and R2FT focused only on the Absorbing approach?  (Absorbing vs uniform)**
>
> The issue you raised can also be framed as a debate between *absorbing (i.e., MDLM, mask diffusion)*  and *uniform* approaches, as discussed in [1]. We anticipate that the absorbing family will remain prevalent, and revocability and dependency may be implemented in a way compatible with absorbing. That is why we concentrated on solving the problem of unrevocable MDLM first (i.e., “third way”). We provide the following rationales.
>
> (1) First, the absorbing-based approach has already proved its downstream task performance comparable to AR LMs (e.g., LLADA). Regardless of its limitations, a method that already works this well is unlikely to be entirely abandoned. In contrast, *uniform* methods have been mainly evaluated on auxiliary metrics such as gen PPL.  We believe this stems from the following theoretical limitations.
>
> (2) Revocation introduces an additional cost.
>
> As the sample quality of diffusion LM is partially a function of sample rate $r=L/S$, revocability increases the number of masks to decode, similar to raising r.  To investigate this, we compared the average unmasking per step (denote r*) of both revocable MDLM (reMDM [2]**)** and irrevocable ours.
>
> ETable1. Average unmasking per step.
>
> | L | S | r*  of reMDM | r* of MDLM |
> | --- | --- | --- | --- |
> | 512 | 65 | 12.98 | 8 |
> | 512 | 128 | 8.69 | 4 |
> | 512 | 256 | 6.668 | 2 |
>
> This shows r* of the revocable model is 2~3 times larger than that of MDLM, equivalent to operating with a step size 2–3 times smaller in MDLM, which causes degradation. This is one of the reasons that the downstream task performance of reMDM is also lower than MDLM (ETable2).
>
> ETable2. AlpacaEval of LLADA-8B-Base based models. L=512, S=128, all CoT instructed, k=5 in topk.
>
> | rft | decode | winrate |  len > 0  | avg Lenght | inlier (95%) |
> | --- | --- | --- | --- | --- | --- |
> |  | categorical | 37.98 | 803 | 360.1 | 1 |
> |  | **remdm** | 41.64 | 778 | 358.1 | 0.96 |
> |  | **remdm** + conv + topk | 42.12 | 805 | 470.5 | 1 |
> |  | topk | 42.48 | 777 | 354.5 | 0.99 |
> |  | conv + topk | 45.27 | 799 | 371.3 | 0.98 |
> | o | categorical | 50.22 | 805 | 403.5 | 1 |
> | o | **remdm** + conv + topk | 50.41 | 805 | 439.9 | 1 |
> | o | **remdm** + topk | 51.68 | 805 | 472.3 | 1 |
> | o | topk | 55.38 | 805 | 459.8 | 1 |
> | o | conv + topk | 67.73 | 805 | 320.1 | 1 |
>
> ETable3. GSM8k accuracy of LLADA-8B-Instruct based models. L=512, S=128, all CoT instructed, k=1.
>
> |  | Acc |
> | --- | --- |
> | remdm + topk | 0.3980 |
> | remdmd + conv | 0.4541 |
> | topk | 0.4200 |
> | conv | 0.5171 |
>
> (3) Revocability may not be a critical property.
>
> As noted in [1], we often consider revocability as an inherent and necessary property of diffusion models (DM), thinking of the image generation service, transforming an existing image into an entirely different one (e.g., a person eating bread → a lion eating bread).
>
> However, a closer look at the denoising process reveals that this is not as flexible as the entire output. To replace a person with a lion, the DM (1) first degrades the original image into an almost unrecognizable noisy state, and (2) then gradually denoise it. In the denoising phase, once strong edges or dots emerge at a certain location in the early steps, these elements tend to remain stable, functioning like anchors. Then, the surrounding blurred regions refine around them, determining how to interpret the anchors (e.g., as an eye or a nose). Thus, while the overall process is *revocable, steps after initial destruction* exhibit limited revocability. This may be related to the fact that the reverse process is more deterministic than explorative, since DDIM [3]. We will show visual examples in the camera-ready.
>
> Likewise, existing successful generation models (e.g., AR) are not revocable. It may be that well-calibrated sampling can provide an enough solution.
>
> (4) If revocability is still needed, a compatible way is promising.
>
> [1] provide theoretical justification for this compatibility. In [1], the diffusion posterior $p(x | x_t)$ can be decomposed as $p(x|x_t) = p(N|x_t)  p(x|x_t, N)$, where N is noise. The “uniform” approach models the entire, which is theoretically complete but practically beyond the model capacity. In contrast, the “absorbing” focuses only on modeling  $p(x|x_t, N)$, which is theoretically incomplete but highly practical.
>
> As an alternative, [1] proposes operating $p(N|x_t)$ independently while preserving absorbing as a main drive, achieving a balance between these two extremes. Remasking (ReMDM) [2] also lies along this stream. As long as absorbing models can operate independently, Conv and R2FT remain fully compatible, as also empirically shown in ETable2, 3.
>
> ---
>
> **[Q-2] R2FT’s rule-based approach is against the Bitter Lesson.  (regarding Question 2)**
>
> (1) In R2FT, the rule-based component is only used for *data construction*, while the learning lies with the model itself. So R2FT is closer to SFT modeling than a rule-based system.
>
> (2) R2FT can be viewed as a low-cost approach to human data annotation.
>
> Creating datasets for desired model behavior has been a guaranteed approach in LLM development. Research fields are further mining new datasets to broaden model coverage. Likewise, we can annotate “undesired” data to prevent unwanted behaviors. However, as “deconstruction is always much simpler than construction**”**, unwanted behaviors can often be easily reproduced through simple rules, which is R2FT.
>
> ---
>
> **[Q-3] When uni/bi directional Conv is used?**
>
> In this work, we employed unidirectional sampling exclusively, as bidirectional sampling did not yield significant performance differences. This is likely because the context is presented only on the left side in QA tasks. We noted this and discussed the possible advantage of policy optimization in the limitation section.
>
> ---
>
> [1] Liu et al. Think while You Generate: Discrete Diffusion with Planned Denoising
>
> [2] Guanghan Wang et al., Remasking Discrete Diffusion Models with Inference-Time Scaling

---

> > ### Comment · Reviewer_TLks · 2025-08-05
> >
> > Thank you for your very thorough response. Despite the length of this reply, I only request your input on two points, clearly identified below in all caps.
> >
> > ---
> >
> > > [...] which explains why patterns in candidate zone, such as “lincoln lincoln lincoln” emerge.
> > >
> > > Such a problem is not a matter of “independently decoding” property.
> >
> > My point is that if you were to unmask a **single** token, say $x_{22}$="lincoln", then call the model again with this $x_{22}$ now unmasked in the model's input, then the model will no longer give high probability for "lincoln" in $x_{21}$ and $x_{23}$. The problem emerges when unmasking **multiple** tokens at the same time: because you're not calling the model again, the three positions are sampled **independently** (and doing so independently).
> >
> > > no model can feasibly consider $|V|^{20}$ possibilities in advance.
> >
> > I fully agree! Thus, your model (like most diffusion models out there) treat each token as independent stochastic variables (conditional on the same shared context). In contrast, [Liu et al. Discrete Copula Diffusion. https://arxiv.org/pdf/2410.01949] capture some (limited) coupling between pairs of tokens: if your neighboring token is "lincoln", it should decrease your probability of being a "lincoln" token yourself.¹
> >
> > To be clear, my claim is that the problem emerges when a single call to the model
> >
> > - samples **multiple** tokens
> > - **independently** and
> > - **irrevocably**.
> >
> > **REQUESTED ACTION: clarify what you mean by '''Such a problem is not a matter of “independently decoding” property.'''** This statement makes me believe that I may not have made my point clear, which would prevent you from properly editing an eventual camera ready to this extent: it is important that we reach agreement on this point.
> >
> >
> > > In this work, we employed unidirectional sampling exclusively, as bidirectional sampling did not yield significant performance differences.
> >
> > I believe that this should be crystal clear in the introduction (and possibly also the abstract).
> >
> > **REQUESTED ACTION: Do you agree to clarify in the introduction of an eventual camera ready that only unidirectional sampling is used?**
> >
> > ---
> >
> > Everything else past this point is just comments: no action required.
> >
> > > This observation might lead you to three questions.
> >
> > Thank you for anticipating these questions.
> >
> > > and revocability and dependency may be implemented in a way compatible with absorbing.
> >
> > I'm with you on this.
> >
> > > (2) Revocation introduces an additional cost.
> >
> > The analysis here considers the "number of unmasking" as the measure of "cost". I would argue that the actual cost comes from the total number of transformer positions that have to be processed, and this depends strongly on the KV caching strategy. When using the caching strategy of [Sahoo et al. Simple and Effective Masked Diffusion Language Models. NeurIPS 2024], then yes: the number of unmaskings rather directly maps to costs. However, there are alternative ways to proceed: you may see [Fathi et al. Unifying Autoregressive and Diffusion-Based Sequence Generation. https://arxiv.org/abs/2504.06416], paying a particular attention to their $\omega$, the number of transformer positions that you need to process each time you call the model.
> >
> >
> > > (1) In R2FT, the rule-based component is only used for *data construction* [...] (2) R2FT can be viewed as a low-cost approach to human data annotation.
> >
> > Yes. R2FT is definitely much better than human annotation. My point is that a human has to be involved in a "meta annotation" step specifying how the data should be constructed.
> >
> > ---
> >
> > ¹: One subtlety: once the model has put two consecutive "lincoln" in the sequence, then adding more "lincoln"s before or after it may actually be in distribution. This is not important: please ignore this footnote if it is more confusing than helping.

---

> > > ### Author Response · Authors · 2025-08-07
> > > **Official Comment by Authors <2>**
> > >
> > > **[Q, REQUESTED ACTION] Do you agree to clarify in the introduction of an eventual camera-ready that only unidirectional sampling is used?**
> > >
> > > = We agree and ensure that these points will be clearly reflected in both the abstract and introduction sections.
> > >
> > > #. In addition, we argue that Conv offers sufficient contribution even without counting bidirectional generation. Conv effectively narrows the decoding window without time-interval expansion problem, distinct from previous SOTA (e.g., semi-AR). This advantage is theoretically guaranteed (in response to reviewer **QCLw**) and also has been empirically validated.
> > >
> > > To clarify this point, we also experimented with the **slide annealing** approach you mentioned. However, we found that its performance severely degraded under the $L = 512, S = 128$ setting. Structural coherence was significantly broken, so AlpacaEval winrate is near zero.
> > >
> > > We suggest the following rationale for this observation:
> > >
> > > The work you referenced [2] appears to mainly assume the case of r=1 (where r=L/S), under which slide annealing can behave like AR. However, when $r$ gets larger, the behavior shifts to resemble autoregressive decoding with a stride of $r$, particularly during early steps when the decoding window is empty. In more intensive settings like ours (r=4 or 8), this leads to very rigid sampling, forcing the unmasking of 4-8 continual tokens at once. This is prone to cause structural incoherence (due to independency), and once such case occurs in early steps, it propagates through later steps, resulting in the breakdown of the entire output.
> > >
> > > Both semi-AR and slide annealing suffer from rigidity, highlighting the value of Conv’s flexibility.
> > >
> > > ---
> > >
> > > **[Q] Revocation introduces an additional cost.**
> > >
> > > = We acknowledge that our previous wording may have caused confusion. We agree with your point regarding the KV cache.
> > >
> > > What we referred to as "cost" was not about computational overhead, but rather about output quality. Specifically, remasking leads to a greater number of tokens to be unmasked per one decoding step. As the number of tokens predicted simultaneously increases, so does the risk of errors due to the “*independency” problem*.
> > >
> > > This is the phenomenon we aimed to show in Tables 1 and 2. In general, outputs with $S=64$ (S is step size) tend to underperform compared to those with $S=128$ (Figure 4), likely due to a higher decoding-per-step ratio (r*). As shown in **ETable 1** of the rebuttal, the r* of reMDM ($S=128$) is higher than that of MDLM ($S=64$). This indicates that reMDM gains revocability, for a cost of a higher r*.
> > >
> > > ---
> > >
> > > **[Q] Regarding R2FT.**
> > >
> > > = We agree with your point. However, we believe this point differs from the type of rule-based systems worried in *The Bitter Lesson*.  R2FT more resembles SFT based on human annotation.
> > >
> > > ---
> > >
> > > [1] Anji Liu et al., DISCRETE COPULA DIFFUSION
> > >
> > > [2] Fathi et al. Unifying Autoregressive and Diffusion-Based Sequence Generation

---

> > > > ### Comment · Reviewer_TLks · 2025-08-07
> > > >
> > > > Thanks again for your reply.
> > > >
> > > > > = It seems the difference in perspective comes from the examples we each assumed; our focus was on the early steps.
> > > >
> > > > No, the copula approach also mitigates the "lincoln lincoln lincoln" issue on the very first time step.
> > > >
> > > > > To state it more formally, consider a model designed to capture dependencies by computing the conditional joint probability
> > > >
> > > > This is exactly the direction taken by [Liu et al.] in their copula model: a first order correction that accounts for pairwise correlations in the tokens to be yielded within the same call to the model. Without copula, the model would predict consecutive "lincoln lincoln" to be likely. The copula biases the distribution so that if either of them is "lincoln", the other one becomes much less likely to be "lincoln".
> > > >
> > > > > In practice, it already incorporates aspects of cross-correlation due to the transformer’s cross-attention mechanism.
> > > >
> > > > To be clear, I fully agree with this statement. My understanding of why you thought I needed to see the above sentence is that you misunderstand what the copula do.
> > > >
> > > > And to further clarify, I agree with your other thoughts about having a separate transformer model: this is a point against copula.
> > > >
> > > > I believe that I now understand your misunderstanding, and your previous answers now make more sense to me in this framing. This is good enough to me: I don't have the time for another iteration before the deadline, so I'll simply strongly encourage you to get a better understanding of [Liu et al.] before editing an eventual camera-ready version.
> > > >
> > > > >> [Q, REQUESTED ACTION] Do you agree to clarify in the introduction of an eventual camera-ready that only unidirectional sampling is used?
> > > >
> > > > > = We agree and ensure that these points will be clearly reflected in both the abstract and introduction sections.
> > > >
> > > > Again, good enough for me.
> > > >
> > > > > To clarify this point, we also experimented with the slide annealing approach you mentioned.
> > > >
> > > > My understanding is that [Fathi et al.] provides the same guarantees as you do, and allows the same kind of flexibility: again, I encourage you to get a good understanding of it before editing an eventual camera-ready. However, they haven't released code, it is still recent work, they don't do any RL, and your consideration of a bidirectional sampling is novel on its own (even as a negative result). I believe that your work is worth publishing.
> > > >
> > > > > = We acknowledge that our previous wording may have caused confusion. We agree with your point regarding the KV cache.
> > > >
> > > > Good.
> > > >
> > > > > = We agree with your point. However, we believe this point differs from the type of rule-based systems worried in The Bitter Lesson. R2FT more resembles SFT based on human annotation.
> > > >
> > > > Yeah, your approach is making "rules" at a higher level, which is definitely a step in the right direction. In fact, automating the creation of low-order rules/data using increasingly-higher order rules may be a valid response to Sutton.
> > > >
> > > > ---
> > > >
> > > > I'm increasing my score to 5.

---

> > > > > ### Author Response · Authors · 2025-08-09
> > > > >
> > > > > We sincerely appreciate the score increase and thank you for engaging in this discussion over an extended period. We will make sure to carefully review the two works you mentioned before preparing the camera-ready version.
> > > > >
> > > > > That said, based on our current understanding, we would like to offer a few additional thoughts below.
> > > > >
> > > > > ---
> > > > >
> > > > > **[#] Regarding Copula, "lincoln lincoln" case.**
> > > > >
> > > > > = We noted in our previous discussion that, even a model that computes the conditional joint probability $p(x_1, x_2 | \tau)$ marginalize to $p(x_1, x_2)$ in early step.
> > > > >
> > > > > Conceptually, $p(x_{21}, x_{22})$ represents token pairs with high internal coherence but lacking contextual grounding. For example, sequences like *“Lincoln Lincoln”* would receive lower probabilities. This partially mitigates our original concern, and **we believe this behavior is consistent with your observation.**
> > > > >
> > > > >  However, meaningless pairs with high prior, such as *“\n \n \n \n”* or *“Response:”* (i.e., *‘Response’* and *‘:’*) could still emerge.  While such tokens could be neutral, they could also be harmful, as early irrelevant tokens may function like an additional instruction prompt, thereby steering the generation into unaligned directions. To test this, we deliberately inserted a meaningless repetitive phrase (e.g., “Response:”) at the right end of the window. This setup was designed to simulate a worst-case scenario in which an meanlingless repetitive phrase is sampled at an early step. As a result, we observed a notable degradation in alignment and structural coherence across the generated outputs (”topk + repetition” in ETable 1).
> > > > >
> > > > > $\color{white}$
> > > > >
> > > > > ETable 1. AlpacaEval of LLADA-8B-Base based models. L=512, S=128, all CoT instructed, k=5 in topk.
> > > > >
> > > > > | RFT | decode | winrate | count | avg Lenght | inlier (95%) |
> > > > > | --- | --- | --- | --- | --- | --- |
> > > > > |  | topk + **repetition** | 36.14 | 780 | 340.56 | 0.95 |
> > > > > |  | topk | 42.48 | 777 | 354.5 | 0.99 |
> > > > > |  | conv | 45.27 | 799 | 371.3 | 0.98 |
> > > > > | o | topk | 55.38 | 805 | 459.8 | 1.00 |
> > > > > | o | conv | 67.73 | 805 | 320.1 | 1.00 |
> > > > >
> > > > > $\color{white}$
> > > > >
> > > > > ---
> > > > >
> > > > > **[#]  Regarding sliding annealing.**
> > > > > = If the “guarantee” you mentioned aligns with what we responded to reviewer **QCLw**, then at a high level, we agree that it holds.
> > > > >
> > > > > However, upon closer inspection, there are important differences. (We assume an implementation on absorbing diffusion.)
> > > > >
> > > > > For our understanding, **the implementation of sliding annealing divides the "active" window  into two distinct regions** (the size of window is denoted as $\omega$ in Figure 3(b) of [1], but we refer to it as $L_b$). This property arises from the fact that the active window must shift by exactly $r=L/S$ positions to the right at each decoding step.
> > > > >
> > > > > (1) **The former $1$ to $r$ positions:**  No masks are allowed to remain in this region after one decoding step. Otherwise, those masked tokens may never be resolved in later steps.
> > > > >
> > > > > (2) **The latter $r + 1$ to  $L_b$  positions:**  They may contain masked tokens after one decoding step.
> > > > >
> > > > > In our theoretical analysis on performance guarantees (Response to QCLw), we assumed that the property of structural coherence degrades as the density $d = \frac{r}{W}$ increases. From this perspective, the part (2) satisfies the guarantee because it operates at a sufficiently low $d$, similar to Conv.
> > > > >
> > > > > In contrast, (1) operates at a much higher $d$, which violates the theoretical guarantee, especially when $r$ gets larger. In particular, during early decoding steps (when few tokens have been unmasked), $d$ reaches its maximum value of 1. This is where the key difference from Conv arises, and where the breakdown of structural coherence is most likely to occur.
> > > > >
> > > > > That said, we could not directly access the implementation of sliding annealing. This method also appears to assume revocability. For instance, under the described procedure, the sample rate becomes greater than $L/S$, which suggests a setting that relies on revocation. (Preventing this required additional hyperparameter tuning, decreasing the stride to lower than $L/S$.)
> > > > >
> > > > > ---
> > > > >
> > > > > In any case, we will take a closer look at both works.
> > > > >
> > > > > We hope to have the opportunity to continue this discussion in the future.
> > > > >
> > > > > Thank you again!
> > > > >
> > > > > ---
> > > > >
> > > > > [1] Fathi et al. Unifying Autoregressive and Diffusion-Based Sequence Generation

---

> ### Author Response · Authors · 2025-08-07
> **Official Comment by Authors <1>**
>
> It's a pleasure to reconnect with you! Thank you for the thoughtful discussion. We marked “REQUESTED ACTION” on the according questions. We are happy to engage with any questions or discussions until the end of the discussion period.
>
> ---
>
> **[Q] The issue arises when sampling multiple tokens at once, not when sampling a single token.**
>
> = We agree. Diffusion models inherently aim to sample multiple tokens to gain speed, which introduces this problem. As noted at the end of Q-1-3, we suggested several ways to mitigate it: (1) increase S (i.e., step size) (2) sampling from only closer positions (e.g., Conv, semi-AR) (3) or introduce revocability.
>
> As you noted, introducing interdependent decoding would also be an effective solution.
>
> ---
>
> **[Q, REQUESTED ACTION] The meaning of ''Such a problem is not a matter of “independently decoding” property.''**
>
> = It seems the difference in perspective comes from the examples we each assumed; our focus was on the early steps.
>
> Case1. Example assumed in the Copula paper [1].
>
> ```
> The <MASK> dog <MASK> the neighbors.
> ```
>
> The example assumed in Copula (**Case1**, from Figure 1 in [1]) involves mid-timesteps, where a substantial amount of masks are already decoded and the remaining possibilities are significantly constrained. **In such cases, we agree that a Copula could serve as an effective solution.**
>
> In contrast, the example we assumed (Q-1-1) is sampling at early steps over an empty space. In this case, the number of variables to consider is extremely high, making it ineffective to precompute dependencies.
>
> To state it more formally, consider a model designed to capture dependencies by computing the conditional joint probability $p(x_{21}, x_{22} | \tau)$, rather than only $p(x_{21} | \tau)$ as in the Q-1-1 example. However, if we assume $\tau$ is in $T$, all possible combination of candidates of masked tokens $\{x_1 \dots x_{20}\}$, then $E_\tau [p(x_{21}, x_{22} | \tau)]$ is marginalized into $p(x_{21}, x_{22})$, from the same mechanism in Q-1-1, leaving no contextual information available. This is what we ment by **''Such a problem is not a matter of “independently decoding” property''**. This necessitates additional solutions to address the concern (e.g., R2FT, or revocability).
>
> ---
>
> **[Q] Could the “Copula” be a promising direction?**
>
> = We agree that if a method effectively captures dependencies, it would be an excellent advance. We also find the Copula [1] highly interesting and a promising direction. Furthermore, its design is compatible with the absorbing approach, making it also compatible with our method. However, we have the following concerns:
>
> Before explaining our concerns, let us first clarify our understanding of the Copula model. While we often simply state that diffusion LMs predict $p(x_t |x_{t+1})$  independently of other positions, this is only partially true. In practice, it already incorporates aspects of cross-correlation due to the transformer’s cross-attention mechanism. Therefore, computing cross-correlations of model outputs (e.g., Copula) can essentially be interpreted as feeding the outputs into an additional transformer-like layer. **Indeed, the implementation of Copula feeds the model outputs into a separate transformer**.
>
> Viewed this way, the following concerns arise:
>
> (1) The conditional probability  $p(c∣\tau)$ ($c$ is candidates) computed by the model indicates the model’s knowledge itself: whether *puppy* is associated with *scared* or *calmed* depends on the distribution of its training data and experience. However, if a separate transformer determines  $p(c∣\tau)$, the final output may become misaligned with the knowledge of the base model.
>
> In this reason, if the Copula model is smaller in size, it may force to generate outputs with less knowledge than the base model. While Copula has demonstrated strong performance in evaluation of structural coherence, it could be not true in downstream tasks requiring extensive knowledge.
>
> (2) The most direct way to address the issue (1) is to scale up the Copula model so that it can absorb sufficient knowledge. However, as the Copula model grows larger, it has little difference from simply doubling the step size.
>
> **This appears to be a critical concern.** In fact, reviewing the Copula paper and its code scripts, we found that a single sampling step requires three separate model inferences (twice for 320M, and once for 137M). Thus, one step of Copula should be compared with 2.5 steps of original model (e.g. S=128 vs S=256–512). However, as you mentioned, increasing the step size alone mitigates much of the dependency issue.
>
> Nevertheless, we believe this approach is highly intriguing and potentially helpful after the mid-step.

---

### Official Review · Reviewer_iU8F · 2025-07-02

**Clarity:** 2
**Significance:** 3
**Originality:** 2
**Rating:** 4
**Confidence:** 3

**Summary:**

This paper identifies the long decoding-window (LDW) problem in diffusion language models, the authors propose Convolutional Decoding and Rejecting Rule-based Fine-Tuning (R2FT) to address this problem. Experiments on benchmarks like AlpacaEval and MT-Bench show their approach achieves state-of-the-art performance among diffusion LMs, enabling efficient generation with fewer decoding steps.

**Questions:**

Most questions/suggestions are reflected in the weakness part, for the last point mentioned before, I suggest the authors conduct a more fine-grained evaluation, analyzing the quality and contextual relevance of generated tokens as a function of their distance from the prompt/context.

One more thing, I was also curious about the performance on bi-directional substream tasks, as mentioned by the authors in the limitations, it would be good if they can provide some basic results.

**Ethical Concerns:**

["NO or VERY MINOR ethics concerns only"]

**Limitations:**

yes

**Quality:**

3

**Strengths And Weaknesses:**

### Strengths
1. This paper systematically analyzes the LDW problem, a core challenge for diffusion LMs, which is a fundamental issue but not fully exploited.
2. The proposed Convolutional Decoding is a simple yet effective, comparing with commonly adopted block/semi-AR methods
3. The R2FT method introduces a targeted, rule-based negative fine-tuning stage, which is also shown to be effective.
4. Results on multiple benchmarks exhibit the potential of the proposed strategies.

### Weakness
1. R2FT relies on simple rule-based negative sample generation, more sophisticated negative mining methods can be considered
2. In most cases,  window size is set to1024(512 in some experiments) , it's hard to judge the robustness/generalization of the method to the window size.
3. The results pure rely on win rate, lacking analyses of more diverse dimensions.
4. It's not clear how exactly the performance improvement comes from, does it primarily stem from scenarios requiring long contextual information?

---

> ### Author Rebuttal · Authors · 2025-07-31
>
> Thank you for the insightful question. We will incorporate these discussions into the camera-ready version.
>
> ---
>
> **[Weakness 1] R2FT relies on simple rule-based negative sample generation; more sophisticated negative mining methods can be considered.**
>
> = The point you raised remains an avenue for future research. However, it can be justified as follows.
>
> (1) We already discussed the strengths of our approach compared to other seemingly more natural alternatives in section 4.2
>
> (2) R2FT can be viewed as a low-cost approach to human data annotation.
>
> Creating datasets for desired model behavior has been a guaranteed approach in LLM development.   Furthermore, the ongoing progress in LLMs relies heavily on continuously generating new datasets to broaden the model coverage. Likewise, we annotate new data for LLMs to prevent unwanted behaviors. However, as “deconstruction is always much simpler than construction**”**, unwanted behaviors can often be easily reproduced through simple rules, which is R2FT.
>
> ---
>
> **[Weakness 4] It's not clear how exactly the performance improvement comes from, does it primarily stem from scenarios requiring long contextual information?**
>
> = If the intent of this question is to inquire about the theoretical guarantees of our method, we would kindly ask you to refer to our response to **QCLw** for further details. We apologize for not including the full details here, as we did not want to occupy excessive space if this was not your intended concern.
>
> ---
>
> **[Weakness 3,4 ] The results are purely based on win rate, lacking analyses of more diverse dimensions. \\ It's not clear how exactly the performance improvement comes from; does it primarily stem from scenarios requiring long contextual information?**
>
> = Acknowledging your concern, we provide an additional metric. Text quality involves two key aspects: alignment and structural coherence. Among these, alignment is primarily assessed using win rate. Besides, s structural coherence is typically assessed using generative perplexity (gen PPL). However, gen PPL is known to overestimate texts containing repeated patterns [1], which makes it easy to be gamed.
>
> Accordingly, we computed the mean ($\mu$) and stds ($\sigma$) of gen PPL from the training dataset, and then measured the proportion of model-generated texts whose PPL falls within the interval $\mu \pm 2\sigma$ (reported as *inlier* in ETable 1). Additionally, we report the *count* of outputs which has zero length before the EOS token, indicating another pattern of structural incoherence.
>
> Furthermore, as you mentioned, our method assumes scenarios where long-form text must be generated, without relying on random unaligned sampling, following recent expectations for LLM-based services.  Thus, we conducted additional experiments under a setting designed to better reflect this assumption. To reflect this setting, we instructed models to generate responses in a Chain-of-Thought format to provide detailed reasoning, and we applied top-5 decoding. As average length increased, we set S=128, which still guarantees around 3x sample rate.
>
> ETable 1. AlpacaEval of LLADA-8B-Base based models. L=512, S=128, all CoT instructed, k=5 in topk.
>
> | RFT | decode | winrate | count | avg Lenght | inlier (95%) |
> | --- | --- | --- | --- | --- | --- |
> |  | semi-AR | 21.5 | 777 | 265.5 | 0.34 |
> |  | categorical | 37.98 | 803 | 360.1 | 1 |
> |  | llada | 39.18 | 805 | 405.1 | 0.75 |
> |  | topk | 42.48 | 777 | 354.5 | 0.99 |
> |  | conv | 45.27 | 799 | 371.3 | 0.98 |
> | o | semi-AR | 26.97 | 805 | 452.56 | 0.80 |
> | o | llada | 47.37 | 805 | 394.8 | 0.82 |
> | o | categorical | 50.22 | 805 | 403.5 | 1.00 |
> | o | topk | 55.38 | 805 | 459.8 | 1.00 |
> | o | conv | 67.73 | 805 | 320.1 | 1.00 |
>
> Results in new setting are consistent with previous one. Furthermore, this multi-dimensional evaluation reveals stronger evidence for our hypothesis about the proposed method, which is explained in the next question.
>
> ---
>
> **[Question 1] I suggest the authors conduct a more fine-grained evaluation, analyzing the quality and contextual relevance of generated tokens as a function of their distance from the prompt/context.**
>
> = Contextual relevance as a function of distance from the prompt is basically illustrated in Figure 3. The probability of candidate $c$ on the given context $x$ is  $p(c|x)=p(x|c) p(c)$, where $p(c)$ represents a frequency without any contextual information. This term corresponds to the high-prior tokens or naive repetition, that we identified as problematic (Section 2.2). Figure 3 visualizes $p(c)$, showing that p(c) peaks at the 5–10 steps away from the context, then remains at a high level thereafter (30–50%). This suggests that the influence of the contextual term $p(x∣c)$ relatively diminishes substantially with increased distance.
>
> R2FT explicitly reduces $p(c)$, which leads to the relative increase of $p(x|c)$, which leads to contextually more aligned sampling.  As shown in **ETable 1,** our multi-dimensional evaluation of alignment (win rate) and structural coherence (inlier) captures these trends well. First, categorical sampling with SFT alone (with high $p(c)$) achieves enough structural coherence (inlier=1.00) but low alignment. More deterministic Topk-k decoding has better alignment, but risks structural degradation. In contrast, R2FT results in a noticeable improvement in both alignment and structural coherence.
>
> ---
>
> **[Weakness 2] In most cases, window size is set to1024(512 in some experiments) , it's hard to judge the robustness/generalization of the method to the window size.**
>
> = Thank you for pointing out this important aspect. We are preparing the experiment and will provide the results during the discussion period if possible.
>
> ---
>
> **[Question 2] One more thing, I was also curious about the performance on bi-directional substream tasks, as mentioned by the authors in the limitations, it would be good if they can provide some basic results.**
>
> =  As you point out, we did not evaluate on bi-directional tasks in our paper. We noted this in the Limitations section and deliberately avoided strongly emphasizing bidirectionality as a major advantage. If you feel that it was overstated, we will further tone down the discussion in the revision.
>
> However, we believe that the bidirectionality introduced by *Conv* offers a highly significant advantage, and we hope future researchers will draw inspiration from this aspect. That is why we made a deliberate effort to mention it, even if only briefly.
>
> In our view, the most impactful application of bidirectionality of Conv lies in tasks that need goal awareness, which requires generation grounding on both previous context and preferred reaction (e.g., negotiation). However, designing such tasks is beyond the current scope, so we leave it for future work.
>
> We also expect  **Decision Transformers (DT) [2]** to be highly synergetic with Conv, which is a policy model with LLM architecture. Current DT predominantly rely on AR models, which are inherently unidirectional. This limitation makes them goal-unaware. As a result, during exploration for RL, they often perform numerous random, goal-agnostic actions, significantly reducing the efficiency of trajectory sampling. In this context, DT with diffusion LM architecture can provide a major breakthrough by enabling more goal-oriented sampling.
>
> However, even in this setting (especially in the long-horizon scenario), bidirectional DTs may still suffer from the LDW problem, and decoding that is grounded in the adjacent context from both sides (start point and goal point) would be particularly advantageous in this setting. It becomes nearly impossible to effectively ground when the trajectory involves multiple intermediate waypoints.  In this scenario, Conv can be a better option than a more rigid semi-AR.
>
> ---
>
> Reference
>
> [1] A. Holtzman et al., The curious case of neural text degeneration
>
> [2] Lil Chen et al., Decision Transformer: Reinforcement Learning via Sequence Modeling

---

> > ### Comment · Reviewer_iU8F · 2025-08-09
> >
> > I appreciate the authors’ rebuttal and will maintain my score, leaning toward acceptance.

---

### Official Review · Reviewer_krtD · 2025-07-03

**Clarity:** 3
**Significance:** 4
**Originality:** 4
**Rating:** 6
**Confidence:** 5

**Summary:**

This paper addresses the long decoding-window problem in diffusion LMs, where distant tokens become irrelevant or repetitive. It proposes Convolutional decoding (Conv) to narrow the window without hard blocks and Rejecting Rule-based Fine-Tuning (R2FT) to align far-away tokens. Experiments on AlpacaEval show better quality and faster decoding with fewer steps.

**Questions:**

Line 135 states that sampling from top-ranked candidates “high chance of selecting only repetition or high-prior tokens positions far from context.” Why? Do high-probability tokens actually cluster near the prefix, as suggested by darker colors in Figure 2?

**Ethical Concerns:**

["NO or VERY MINOR ethics concerns only"]

**Limitations:**

Yes

**Quality:**

4

**Strengths And Weaknesses:**

**Strengths**

1. Block decoding sacrifices speed and bidirectionality, but R2FT with convolutional decoding overcomes this trade-off.
2. The method better aligns tokens far from context, yielding faster decoding and improved results.
3. Evaluating on AlpacaEval instead of only NLU tasks better reflects open-ended generation performance.
4. Decoding-window analysis is clear and offers valuable insight.

**Weaknesses**

1. Claim that semi-AR sacrifices speed may not hold when KV-cache is used in diffusion models.
2. All analyses in Section3 use relatively small MDLMs. It’s unclear if results hold for larger models (e.g., 7B LLaDA).
3. Convolution direction needs to be clarified. Uni-directional convolution mimics causal decoding in AR models. Which convolution type is used in the end, uni- or bi-directional?

**Writing suggestions**

1. Intro detail: the motivation and goal of methods are introduced, but without design details. Adding a brief overview of how Conv and R2FT work could be better.
2. Figure 4: Zoom into x=[256,512,1024], y=[0,1000] or use log-scale for PPL to improve readability.
3. Citations: the following key related works on diffusion language models are not cited.
- A Reparameterized Discrete Diffusion Model for Text Generation. https://openreview.net/pdf?id=PEQFHRUFca
- Scaling Diffusion Language Models via Adaptation from Autoregressive Models. https://openreview.net/forum?id=j1tSLYKwg8

---

> ### Author Rebuttal · Authors · 2025-07-31
>
> We sincerely appreciate your positive assessment of the strengths of our work. We will incorporate all of the discussions below into the camera-ready version, along with your suggestions on citations and writing.
>
> ---
>
> **[Weakness 1] Claim that semi-AR sacrifices speed may not hold when KV-cache is used in diffusion models.**
>
> = We agree with this point.  However, the speed gains from KV-cache in semi-AR models are compensated by the increased cost from timespan expansion, implying that diffusion LMs cannot fully leverage their inherent speed advantage with semi-AR.
>
> In contrast, Conv can also theoretically leverage KV-cache, as the boundary of the mask tokens shifts consistently, as illustrated in Figure 7. For this reason, Conv can be considered to have a speed advantage.
>
> ---
>
> **[Question 1] Line 135 states that sampling from top-ranked candidates “high chance of selecting only repetition or high-prior tokens positions far from context.” Why? Do high-probability tokens actually cluster near the prefix, as suggested by darker colors in Figure 2?**
>
> = A key characteristic of diffusion LMs is decoding tokens that are far from the immediate context (LDW problem). To formalize this, consider a scenario in which the model decodes  $x_{21}, x_{22}, x_{23}$  at the first sampling step,  where $x_i$ denote a mask token that is i-th position from given context. The probability of candidate $c$ for $x_{21}$ is  $p(c|\tau)=E_\tau [ p(\tau|c)p(c)]$, where $\tau \in T$  is all possible combination of $\{x1,x2,...,x20\}$. The cardinality of T is $|V|^{20}$ (V is vocab size), the infinity. Therefore, the posterior $p(c \mid \tau)$ converges to $p(c)$. This is consistent with the empirical observation in Figure 3, where the probability of high-prior or repetitive tokens increases and then plateaus at a high level as the distance from the given context grows.
>
> $p(c)$ represents the probability of a candidate that can be easily predicted without any contextual information. Typical examples include high-prior tokens (e.g., function words) or simple repetition of the given context. The same applies to $x_{22}$ and $x_{23}$. Consequently, as illustrated in the upper part of Figure 11, top candidates often include patterns such as “:  :  :” (repetition) or “the the the” (high-prior tokens).
>
> The motivation of R2FT is: if guessing is inevitable, let it be at least more aligned. Formally, this is training the model to reduce $p(c)$ in  $E[p(\tau|c)p(c)]$. By doing so, the probability mass shifts toward tokens with higher average likelihood within $T$. In other words, we choose the candidate patterns like "president president president", which at least contains more contextual information than ": : :".  This is consistent with our empirical observation on AlpacaEval that R2FT achieves better alignment with the query.
>
> ---
>
> **[Weakness 2]** All analyses in Section3 use relatively small MDLMs. It’s unclear if results hold for larger models (e.g., 7B LLaDA).
>
> = That is a valid point. We are preparing the experiment and will share the results during the discussion period.
>
> ---
>
> **[Weakness 3]** Convolution direction needs to be clarified. Uni-directional convolution mimics causal decoding in AR models. Which convolution type is used in the end, uni- or bi-directional?
>
> = The bidirectional decoding we referred to applies to scenarios where context is provided on both sides (left and right), unlike simple QA settings, and we did not evaluate on such tasks in our paper. We noted this in the Limitations section and deliberately avoided strongly emphasizing bidirectionality as a major advantage. If you feel that it was overstated, we will further tone down the discussion in the revision.
>
> However, we believe that the bidirectionality introduced by *Conv* offers a highly significant advantage, and we hope future researchers will draw inspiration from this aspect. That is why we made a deliberate effort to mention it, even if only briefly.
>
> In our view, the most impactful application of bidirectionality of Conv lies in tasks that need goal awareness, which requires generation grounding on both previous context and preferred reaction (e.g., negotiation). However, designing such tasks is beyond the current scope, so we leave it for future work.
>
> We also expect  **Decision Transformers (DT) [2]** to be highly synergetic with Conv, which is a policy model with LLM architecture. Current DT predominantly rely on AR models, which are inherently unidirectional. This limitation makes them goal-unaware. As a result, during exploration for RL, they often perform numerous random, goal-agnostic actions, significantly reducing the efficiency of trajectory sampling. In this context, DT with diffusion LM architecture can provide a major breakthrough by enabling more goal-oriented sampling.
>
> However, even in this setting (especially in the long-horizon scenario), bidirectional DTs may still suffer from the LDW problem, and decoding that is grounded in the adjacent context from both sides (start point and goal point) would be particularly advantageous in this setting. It becomes nearly impossible to effectively ground when the trajectory involves multiple intermediate waypoints.  In this scenario, Conv can be a better option than a more rigid semi-AR.
>
> ---
>
> References
>
> [1] Lil Chen et al., Decision Transformer: Reinforcement Learning via Sequence Modeling NeurIPS 2021

---

> > ### Comment · Reviewer_krtD · 2025-08-04
> >
> > Thank you for your clarification. After reading your responses to me and the other reviewers, I noticed that in most of your experiments you actually use a unidirectional convolutional setup. This raises my concern and may lead me to reconsider my review score.
> >
> > Specifically, mentioning bidirectional convolution in the paper is misleading, as it suggests you are using a true bidirectional model when in fact it appears only as a brief note on future work. In your experiments, the unidirectional convolution behaves much like forced causal decoding, essentially moving toward AR decoding and therefore straying from the original intent of diffusion models for text generation.

---

> ### Author Response · Authors · 2025-08-04
>
> Thank you for the careful review. We would like to offer the following clarification.
>
> ---
>
> **(1) Diffusion LM has bidirectional attention.**
>
> To provide a clearer explanation, we distinguish between “*bidirectional attention”* and “*bidirectional generation”*.
>
> AR models inherently employ **unidirectional attention** in their architecture, whereas diffusion-based LM architectures leverage **bidirectional attention**, which applies to both Conv and semi-AR.
>
> Therefore, even when diffusion LMs perform **unidirectional generation**, they continue to maintain **locally bidirectional attention**. This applies to both semi-AR and Conv variants.
>
> For example, consider the prompt:
>
> *“Who is the president: __________”*
>
> If, in the next step, a token happens to be sampled in the middle, resulting in:
>
> *“Who is the president: ______is __”*,
>
> then the masked positions between “*president:”* and “*is”* will attend to context on both sides (e.g., “president:”, “is”). Thus, bidirectional attention occurs in a **local level.**
>
> This behavior can be observed in Figure 7(a), where sampling does not proceed strictly sequentially as in AR but occurs in a scattered manner. The mask tokens between scattered tokens are applied with bidirectional attention. This bidirectionality is a difference from AR models.
>
> However, this situation (activating bidirectional attention) is largely a byproduct of the model sampling multiple positions in a scattered manner, **which might provide little benefit in most of the existing LLM tasks.**  This is because, in most LLM tasks (e.g., QA), useful information is mainly provided on the left side of the window. In fact, this property can even introduce the LDW problem, where tokens sampled far from the given context become misaligned, forcing harmful bidirectional attention.
>
>  $\color{white}$
>
> **(2) Conv is capable of bidirectional generation.**
>
> “Bidirectional generation” refers to the overall direction in which generation is globally grounded. Semi-AR enforces a strict rule to proceed from left to right, whereas Conv, when context is provided on both sides, automatically grounds generation multi-directionally. This distinction is illustrated in Figure 7.
>
> Tasks that provide context on both sides are **goal-oriented**, as explained. For example, negotiation requires aligning with previous utterances and desired responses. And even in QA tasks, desired responses or scores can be framed as explicit goals. This naturally offers a way to reinforce LLMs toward alignment with annotated preferences. **This is also the scenario where the bidirectional attention of diffusion LMs provides a clear benefit.**
>
> However, as noted in our limitations, such bi-context tasks are currently rare—likely because most LLM tasks have historically been designed for AR-based models. If diffusion LMs gain broader adoption, we anticipate that more of these tasks will emerge.
>
> Therefore, we believe that solving such tasks represents one of the key motivations behind the growing interest in diffusion LMs, which is why we chose to report the bidirectional generation property of Conv.
>
> Of course, this claim has not been empirically validated due to the absence of such tasks. We have noted this as a limitation and believe that explaining the underlying principles of Conv sufficiently conveyed our setting without misleading. To avoid any misleading, we will emphasize this point more clearly in the revised version.
>
>  $\color{white}$
>
> **#. Why is Conv valuable, and how does it differ from AR?**
>
> Without bidirectional generation, Conv still offers a superior approach compared to the previous SOTA method (semi-AR) for addressing the LDW problem, **which is our main claim for contribution.**
>
> As you noted, Conv enforces a generation process grounded in the provided context, similar to AR. However, this mechanism serves a specific purpose: mitigating the LDW issue.
>
> Default diffusion LMs often generate tokens at distant positions, leading to outputs that are poorly aligned with the given context (Section 2). In contrast, AR avoids LDW because it only unmasks the position attached to the context. Semi-AR, the previous SOTA method, mimics this benefit of AR by strictly partitioning the decoding space into blocks and processing them left-to-right (i.e., autoregressively). Yet, this approach introduces a time-interval expansion problem, negating the speed advantage over AR.
>
> Conv, by comparison, narrows the context to reduce LDW like semi-AR, while avoiding the time-interval expansion issue, as we provided both theoretical guarantees (Response to reviewer **QCLw**) and empirical evidences (Figure8, 9). As a result, Conv mitigates LDW without sacrificing the decoding speed advantage, **resulting in faster speed than AR**—a key distinction from both AR and semi-AR.

---

### Official Review · Reviewer_QCLw · 2025-07-05

**Clarity:** 4
**Significance:** 3
**Originality:** 3
**Rating:** 5
**Confidence:** 4

**Summary:**

The paper introduces advancements in diffusion-based language models (LMs) to address the long decoding-window (LDW) problem, where tokens generated far from the input context become irrelevant or repetitive. Unlike autoregressive (AR) models, diffusion LMs can generate multiple tokens in parallel and leverage bidirectional context, but their performance suffers from the LDW issue. The submission proposes two key methods to mitigate this:

- Convolutional Decoding (Conv): A normalization technique that narrows the decoding window without segmenting it into blocks, preserving both decoding speed and bidirectionality. This contrasts with semi-autoregressive (semi-AR) approaches, which sacrifice these advantages.
- Rejecting Rule-based Fine-Tuning (R2FT): A post-hoc training method that reduces the model's tendency to produce repetitive or high-prior tokens by aligning distant tokens with the context, enhancing fluency and coherence.

In addition, the submission introduces EOS-fill, a technique that improves decoding speed by filling positions with end-of-sentence tokens when one is sampled. The proposed methods were evaluated on open-ended text generation benchmarks like AlpacaEval, MT-Bench, and Wiki, using the Masked Diffusion Language Model (MDLM) backbone. The results show state-of-the-art performance among diffusion LM baselines, achieving high-quality text generation with significantly fewer decoding steps compared to prior work.

**Questions:**

- The submission provides response samples but lacks a detailed discussion of scenarios where Convolutional Decoding (Conv) or Rejecting Rule-based Fine-Tuning (R2FT) fail or produce suboptimal outputs. Could you include a dedicated section or table analyzing specific failure cases (e.g., types of prompts or contexts where repetition persists or coherence degrades)?

- The submission lacks theoretical analysis or guarantees for why Conv avoids the time-interval expansion problem or why R2FT effectively mitigates repetition. Could you provide a theoretical explanation (e.g., a simplified model or derivation) of how Conv’s normalization preserves decoding quality compared to semi-AR, or how R2FT’s loss function reduces high-prior token preference?

- The submission specifies GPU types (NVIDIA RTX A5000) and training steps but omits details on execution time, memory usage, or total compute cost for training and inference. Could you provide a table or paragraph detailing these resources (e.g., hours for SFT/R2FT, VRAM per GPU, total GPU-hours) for both small (18M) and large (8B) models?

**Ethical Concerns:**

["NO or VERY MINOR ethics concerns only"]

**Final Justification:**

Congratulations on the submission. Excellent paper. Hope it gets accepted.

**Limitations:**

yes

**Paper Formatting Concerns:**

No concern

**Quality:**

4

**Strengths And Weaknesses:**

1. Quality

*** Strengths ***

*Rigorous Experimental Design*: The submission conducts comprehensive experiments using established benchmarks (AlpacaEval, MT-Bench, Wiki, Arena-Hard-Auto, GSM8K) to evaluate the proposed methods (Convolutional Decoding, R2FT, EOS-fill). It compares against strong baselines, including MDLM with categorical sampling, LLADA, and top-k global normalization, ensuring a robust evaluation of performance improvements. For example, Table 1 highlights AlpacaEval results, with clear reporting of first and second-best performances.

*Reproducibility*: The submission provides detailed experimental settings, including decoding window size ($ L = 1024 $), step size ($ S = 128 $), kernel/block sizes, model sizes (18M and 8B), and training protocols (SFT, R2FT, LoRA). The inclusion of anonymized code and a URL for supplementary material enhances reproducibility. Hyperparameters, such as learning rate ($ 3e-5 $), batch size (512), and optimizer (AdamW), are explicitly stated.

*Statistical Rigor*: The paper reports error bars using standard deviations for AlpacaEval, adhering to the official evaluation interface. This allows for a reliable assessment of variability in win rates across decoding strategies, strengthening the statistical significance of the results.

*Theoretical Grounding*: The submission builds on the simplified NELBO objective for diffusion LMs and provide a clear explanation of the LDW problem using statistical observations. The mathematical formulation of R2FT’s training objective is well-defined, drawing from established methods like DPO and SimPO.

*Thorough Baselines*: The submission evaluates multiple decoding strategies (categorical, LLADA, top-k global, Conv) and includes both small (18M) and large (8B) models, ensuring a comprehensive comparison. The use of the Alpaca instruction dataset for fine-tuning and the detailed setup for SFT and R2FT further strengthens the experimental quality.

*** Weaknesses ***

*Compute Resource Details*: Although the submission specifies GPU types (NVIDIA RTX A5000) and training steps, it lacks precise information on execution time or memory requirements for training and inference.

2. Clarity

*** Strengths ***

*Clear Problem Definition*: The submission clearly articulates the long decoding-window (LDW) problem, supported by intuitive explanations and visualizations. The breakdown of token types (repetition, high-prior, meaning, noise) is accessible and well-illustrated.

*** Weaknesses ***

*Overuse of Acronyms*: The submission uses multiple acronyms (MDLM, LDW, R2FT, SFT, Conv, LLADA) without always redefining them in later sections. This could confuse readers who skip sections or are less familiar with the domain.

3. Significance

*** Strengths ***

*Addressing a Key Limitation*: The LDW problem is a significant barrier to the practical adoption of diffusion LMs, which promise faster and bidirectional decoding compared to AR models. By tackling this issue, the submission advances the field toward more viable non-autoregressive language models.

*Practical Impact*: The proposed methods (Conv, R2FT, EOS-fill) achieve state-of-the-art performance among diffusion LMs with a reduced step size ($ S = 128 $). This improves decoding speed, making diffusion LMs more competitive with AR models for real-world applications like chatbots or content generation.

*Broad Applicability*: The techniques are model-agnostic within the MDLM framework and could potentially be adapted to other diffusion-based architectures. The use of the Alpaca instruction dataset and benchmarks like MT-Bench and Wiki ensures relevance to practical NLP tasks.

*Advancing Diffusion LMs*: The submission builds on foundational work (e.g., D3PM, LLADA) and addresses critical limitations of semi-AR decoding (speed and bidirectionality loss). This positions diffusion LMs as a stronger alternative to AR models, potentially influencing future research in non-autoregressive generation.

4. Originality

*** Strengths ***

*Novel Methods*: The proposed Convolutional Decoding (Conv) and Rejecting Rule-based Fine-Tuning (R2FT) are original contributions. Conv introduces a normalization-based approach to narrow the decoding window without sacrificing bidirectionality, distinct from semi-AR’s block-based segmentation. R2FT is a novel post-hoc training strategy that mitigates repetition and high-prior tokens without expensive human annotations, unlike RLHF.

*EOS-fill Innovation*: The EOS-fill technique is a creative optimization that leverages caching to accelerate decoding by filling positions with EOS tokens, a method not widely explored in prior diffusion LM work.

*** Weaknesses ***

*Incremental over LLADA*: While Conv and R2FT are novel, they build heavily on the LLADA framework and MDLM backbone. The core idea of narrowing the decoding window is inspired by semi-AR decoding, which reduces the perceived novelty of Conv. A clearer differentiation from LLADA’s semi-AR approach could strengthen the originality claim.

---

> ### Author Rebuttal · Authors · 2025-07-31
>
> We greatly appreciate the reviewer’s acknowledgment of our work’s strengths and the insightful questions that contribute to its robustness. These discussions will be incorporated into the camera-ready submission.
>
> ---
>
> **[Question2-1] Theoretical guarantee for why Conv avoids the time-interval expansion problem.**
>
> = We provide two theoretical explanations for why Conv may outperform semi-AR. The following discussion assumes a mask diffusion type [1].
>
> **(1) Violation of the training assumption.**
>
> This point is discussed in detail in Section 3.2 (line 174).
>
> **(2) Explanation with the hazard function.**
>
> A more intuitive explanation is that Conv is more flexible than semi-AR, granting the model greater freedom. This can be theoretically justified using the hazard function, which measures the probability of corruption.
>
> Let us follow the notation in the paper, line 174: L as window size, S as step size, b as the number of blocks, $L_b$ as the block size or kernel size, and $S_b$ as the number of steps assigned for each block in semi-AR.
>
> We newly define the sample rate $r=L/S$, and $W$ as the number of mask tokens that the model have to sample from at current step. For example for default diffusion LM, $W=L$ holds at the first decoding step, and $W \approx L-r$ at the second step, since the model might have unmasked $r$ masks at previous step, on average. For semi-AR, it is $L_b$ to $L_b - r$.
>
> We further define the hazard function $P$, representing the probability (or degree) that the final decoded text becomes structurally (grammatically, syntactically) harmed. Conversely, $1-P$ representes the probability of being unharmed (i.e., structurally coherent). We define $Q = \log(1-P)$. And let small $p_t$ denote the probability that the final output becomes harmed due to the decoding at intermediate timestep $t$. Here, the hazard function is commonly defined as $P= 1- \prod_{t=1}^S (1-p_t)$, which in turn $Q= \sum_{t=1}^S \log(1-p_t)$. We assume that $p_t$ depends on both $r$ and $W$, thus we denote $p_t=p_r(W)$.
>
> **#. Assumptions about the factor in the quality of a text.**
>
> We assume that the quality of generated text is related to **density of decoding** (denoted as ****$d=r/W$) in two opposing ways (though there can be more other factors). Raising $d$ can be benefit and harm the quality at the same time.
>
> (1) Alignment:  When the positions to be decoded are far from the given context, the model tends to suggest less aligned candidates—a phenomenon referred to as the LDW problem**.** This situation is often associated with low $d$ (though not always), which indicates lots of masked positions are between the given context and the targeted position  (refer to response of **Question2-2)**. A common strategy to mitigate this issue is to increase $d$  by reducing $W$ (e.g., semi-AR, Conv).
>
> (2) Structural coherence: Conversely, when $d = \frac{r}{W}$ becomes too high, it can reduce the dependency among decoded tokens and lead to structural degradation. This effect stems from a key property of mask diffusion: multiple tokens generated simultaneously cannot attend to one another at the output layer, resulting in low inter-token dependency. Consequently, when many tokens are generated within a narrow span, the likelihood of syntactic conflicts increases.
>
> However, when the distance between two generated tokens $x_i$ and $x_j$ is large “enough”, even if their immediate dependency is weak, the number of possible connecting sequences between them grows exponentially as $|V|^{(j-i)}$ (V: vocab size). This implies that later decoding steps have the opportunity to restore structural connectivity. This is why $p_t$ depends on $r$ and $W$. If large r and small W mean too packed generation, thus low $p_t$.
>
> We formalize this property as follows:
>
> $\text{if } r_1 = r_2 \text{ and } W_1 > W_2, \text{ then } p_{r_1}(W_1) \leq p_{r_2}(W_2)$
>
> Equality approximately holds when $r$ is sufficiently small or when both $W_1, W_2$ are sufficiently large over some threshold. The precise threshold for “sufficiently” and the shape of the curve $p_r(W)$ as a function of $W$ may vary across models.
>
> **#. Comparison Across Decoding Types**
>
> From the two factors discussed earlier, we now focus on **(2) structural robustness** and model it using the hazard function across different decoding strategies, leaving (1) aside for simplicity.
>
> 1. Default mask modeling
>
> For a standard mask diffusion model without a narrowed decoding window, the hazard function $Q_0$ is as follows.
>
> $P \approx 1-\Pi_{t=1}^{S}(1-p_r(L-(t-1) \cdot r))=1-\Pi_t^{S}(1-p_r(t \cdot r))$
>
> $Q_0 = \sum_t^S q_r(t \cdot r)$     ,   $q = log(1-p)$  ,  $W_t = t\cdot r \leq L$
>
> $q_r(W)$ represents the log probability that decoding at a step does **not** introduce corruption. $W$ decreases gradually by $r$ per step.
>
> 1. semi-AR
>
> For semi-AR, when $L$ is divided into 4 blocks (i.e., $L_b = 128$):
>
> $Q_{sa}=b*\sum_t^{S_b}q_r( t \cdot r)$ *,     $W_t = t \cdot r \leq L_b$*
>
> To compare, since $S > S_b$, the value of $q_r(W_t)$ for semi-AR at any given step is strictly smaller than that of the default setting (because $W_t \leq L_b \leq L$  for semi-AR).
>
> Therefore:  $Q_{sa} \leq Q_0$
>
> Equality occurs when the block size $L_b$ approaches $L$, or when $S$ is large enough so that the argument of $p_r(W)$ exceeds its threshold for most steps. This theoretical behavior aligns with the empirical observation in Figure 4, where evaluated structural quality increase as $L_b$ or $S$ are raised.
>
> 1. Conv
>
> For Conv, assume the kernel size $L_b=128$. In this case, at every step we have $W\geq 128$.  Although the actual variation of $W$ is more complex, for simplicity, we assume $W = 128$.
>
> $Q_c = (L-128) * q_r(128)  +  \sum_t^{128/r}q(t \cdot r)$
>
> Here, the second term is shared with semi-AR and the default. but the first term is always larger than that of semi-AR, except at the start point of each block.
>
> Therefore, $Q_{sa} \geq Q_b$
>
> As before, equality approximately holds when $S$ is sufficiently large.
>
> Compared to the default mask diffusion, the first term is always smaller than the default, so we have: $Q_c \leq Q_0$.
>
>  Equality holds when the fixed kernel size is “sufficiently” large, so that the effective $W$ go over threshold. This behavior is consistent with the empirical observation shown in Figure8, where kernel size over some threshold yields structural stableness.
>
> To wrap up, In terms of structural robustness, we generally observe the ordering:
>
> $\text{semi-AR} \prec \text{conv}\prec \text{default}.$
>
> However, from the perspective of alignment, the relationship reverses:
>
> $\text{semi-AR}, \text{conv} \succ \text{default}.$
>
> Consequently, adopting Conv offers a favorable trade-off, as it provides substantial gains in alignment while maintaining significantly better structural robustness compared to semi-AR.
>
> This argument is consistent with our empirical findings. Categorical sampling has good structural coherence (inlier in ETable1), while alignment (winrate) degrades. Conv has better alignment and winrate, while semi-AR fails in both.
>
> ETable 1. AlpacaEval of LLADA-8B-Base based models. L=512, S=128, all CoT instructed, k=5 in topk.
>
> | RFT | decode | winrate | count | avg Lenght | inlier (95%) |
> | --- | --- | --- | --- | --- | --- |
> |  | semi-AR | 21.5 | 777 | 265.5 | 0.34 |
> |  | categorical | 37.98 | 803 | 360.1 | 1 |
> |  | topk | 42.48 | 777 | 354.5 | 0.99 |
> |  | conv | 45.27 | 799 | 371.3 | 0.98 |
> | o | topk | 55.38 | 805 | 459.8 | 1.00 |
> | o | conv | 67.73 | 805 | 320.1 | 1.00 |
>
> ---
>
> **[Question2-2] Theoretical guarantee for why R2FT effectively mitigates.**
>
> =We divide the explanation into two steps: the decoding step and the training step.
>
> (1) Decoding step
>
> First, let us describe the scenario where the issue arises. A key characteristic of diffusion LMs is decoding tokens that are far from the immediate context (LDW problem). To formalize this, consider a scenario in which the model decodes  $x_{21}, x_{22}, x_{23}$  at the first sampling step,  where $x_i$ denote a mask token that is i-th position from given context. The probability of candidate $c$ for $x_{21}$ is  $p(c|\tau)=E_\tau [ p(\tau|c)p(c)]$, where $\tau \in T$  is all possible combinations of $\{x1,x2,...,x20\}$. The cardinality of T is $|V|^{20}$ (V is vocab size), infinity. Therefore, the posterior $p(c \mid \tau)$ converges to $p(c)$.
>
> p(c) represents the probability of a candidate that can be easily predicted without any contextual information. Typical examples include high-prior tokens (e.g., function words) or simple repetition of given context. The same applies to $x_{22}$ and $x_{23}$. Consequently, as illustrated in the upper part of Figure 11, top candidates often include patterns such as “: : :” (repetition) or “the the the” (high-prior tokens).
>
> The motivation of R2FT is: if guessing is inevitable, let it be at least more aligned. Formally, this is training the model to reduce $p(c)$ in  $E[p(\tau|c)p(c)]$. By doing so, the probability mass shifts toward tokens with higher average likelihood within $T$. In other words, we choose the candidate patterns like "president president president", which at least contains more contextual information than ": : :".  This is consistent with our empirical observation on AlpacaEval that R2FT achieves better alignment with the query.
>
> (2) Training step
>
> Rationale on reducing p(c) is described in Section 4.2
>
> ---
>
> **[Question 1] Failure cases**
>
> (1) Conv: As discussed above, Conv exhibits reduced structural coherence when the kernel size or the step size becomes excessively small.
>
> (2) R2FT: Without incorporating the NELBO term during training, R2FT tends to eliminate high-prior tokens entirely. However, this issue is resolved by adding the NELBO term and setting $\gamma = 0.1$.
>
> Such cases will be included in the camera-ready version.
>
> ---
>
> **[Question 3] GPU usage and time**
>
> We will include in the camera-ready version or report in the discussion period.

---

> > ### Comment · Reviewer_QCLw · 2025-08-05
> > **no revision**
> >
> > Congratulations on a very strong paper. I hope this gets accepted.

---

> > > ### Author Response · Authors · 2025-08-08
> > >
> > > We sincerely appreciate your careful consideration of our response, as well as the score increase. We will ensure that this discussion, along with all of your suggestions, is carefully incorporated in the camera-ready version.

---

### Official Review · Reviewer_7TXB · 2025-07-07

**Clarity:** 3
**Significance:** 4
**Originality:** 3
**Rating:** 4
**Confidence:** 3

**Summary:**

This paper works on diffusion-based language models and the long decoding-window problem. The authors propose two new techniques: Convolutional Decoding and Rejective Fine-Tuning. These methods improve speed, and bidirectionality without block-wise compromises. The paper provides strong empirical results on AlpacaEval and other benchmarks.

**Questions:**

-- Can Conv methid generalize to structured generation tasks (e.g., code or math)?
-- Could R2FT interfere with learning domain-specific high-prior tokens?
-- How sensitive are the methods to kernel size or corruption rules in R2FT?

**Ethical Concerns:**

["NO or VERY MINOR ethics concerns only"]

**Final Justification:**

Thanks again for the hard work and well executed response.
I maintain my score due not limited evaluation and benefits of bidirectionality.

**Limitations:**

yes

**Paper Formatting Concerns:**

/

**Quality:**

4

**Strengths And Weaknesses:**

Strengths:
-- Clear motivation and problem formulation.
-- Proposes simple, effective solutions that do not need architectural changes.
-- Strong experimental results, especially in open-ended generation tasks.

Weaknesses:
-- Paper is dense and somewhat hard to follow in parts. Lots can be done here to make it easier to follow and less dense.
-- Weak investigation of bidirectional downstream tasks, despite bidirectionality being a central claim.
-- Some response samples still show factual or stylistic oddities.
-- Evaluation on tasks like reasoning or structured output is limited.
-- Paper is too packed. Figures are too small. This limits readability and potential impact of this work.

---

> ### Author Rebuttal · Authors · 2025-07-31
>
> Thank you for raising such valuable questions that make our work more robust. We will incorporate all related discussions into the camera-ready version.
>
> ---
>
> **[Q1] The paper is dense and somewhat hard to follow in parts. Lots can be done here to make it easier to follow and less dense. Paper is too packed. Figures are too small. This limits readability and potential impact of this work.**
>
> = We appreciate your observation. The formatting likely became dense as we attempted to cover all the necessary details. We will improve readability in the camera-ready version.
>
> ---
>
> **[Q2]  Some response samples still show factual or stylistic oddities.**
>
> = That is correct. While our method improves performance over previous approaches, diffusion LMs still face several inherent limitations—most notably, sensitivity to the step size ($S$) and the lack of revocability. The latter issue has been partially addressed by recent work such as reMDM [1], which introduces remasking strategies, that are also compatible with our approach. Exploring such limitation remains an important direction for future research.
>
> ---
>
> **[Q3] Evaluation on tasks like reasoning or structured output is limited.  Can Conv methid generalize to structured generation tasks?**
>
> How sensitive are the methods to kernel size?
>
> =  Conv generalize to structured generation tasks, significantly outperforming the competitive baseline LLADA on reasoning tasks (GSM8K, MMLU) (ETable 1-2). And Conv also exhibits minimal sensitivity to kernel size even in small step size ($S = 64$).
>
> ETable 1. GSM8k with LLADA-8B-instruct  ,   L=512,  S=64,  CoT instruct.  Each column  is kernel size or block size
>
> |  | 32 | 64 | 128 | 256 | 512 |
> | --- | --- | --- | --- | --- | --- |
> | LLADA | 27.22 | 31.24 | 32.75 | 23.73 | 29.64 |
> | Conv | 47.61 | 44.28 | 40.33 | 39.27 | 39.73 |
>
> ETable 2. MMLU with LLADA-8B-instruct  ,   L=512,  S=64,  CoT instruct.  Each column  is kernel size or block size
>
> |  | 32 | 64 | 128 | 256 | 512 |
> | --- | --- | --- | --- | --- | --- |
> | LLADA | 9.63 | 14.46 | 17.98 | 15.68 | 26.41 |
> | Conv | 44.21 | 46.91 | 50.13 | 51.78 | 54.90 |
>
> ---
>
> **[Q4] How sensitive is the R2FT to the corruption rule?**
>
> = As shown in Algorithm 1, the corruption rule conducts random sampling based on a few predefined rules. Consequently, slight variations of hyperparameters in these rules appeared to have minimal impact on performance, and we were able to reach strong results without extensive tuning of these rules.
>
> However, the training objective shows more sensitivity to the hyperparameters $\gamma$ and $\beta$. Excessive unlearning of unpreferred cases can lead to the elimination of high prior tokens, thereby removing essential functioning words (e.g., *is*, *a*). This issue appears to stem from the nature of the unlearning process itself. Nevertheless, we addressed it by incorporating an additional objective term $\mathcal{L}_{NELBO}$ and setting $\gamma = 0.1$, which mitigated this problem effectively.
>
> ---
>
> **[Q5] Weak investigation of bidirectional downstream tasks, despite bidirectionality being a central claim.**
>
> =As you point out, we did not evaluate on bi-directional tasks in our paper. We noted this in the Limitations section and deliberately avoided strongly emphasizing bidirectionality as a major advantage. If you feel that it was overstated, we will further tone down the discussion in the revision.
>
> However, we believe that the bidirectionality introduced by *Conv* offers a highly significant advantage, and we hope future researchers will draw inspiration from this aspect. That is why we made a deliberate effort to mention it, even if only briefly.
>
> In our view, the most impactful application of bidirectionality of Conv lies in tasks that need goal awareness, which requires generation grounding on both previous context and preferred reaction (e.g., negotiation). However, designing such tasks is beyond the current scope, so we leave it for future work.
>
> We also expect Decision Transformers (DT) [2] to be highly synergetic with Conv, which is a policy model with LLM architecture. Current DT predominantly rely on AR models, which are inherently unidirectional. This limitation makes them goal-unaware. As a result, during exploration for RL, they often perform numerous random, goal-agnostic actions, significantly reducing the efficiency of trajectory sampling. In this context, DT with diffusion LM architecture can provide a major breakthrough by enabling more goal-oriented sampling.
>
> However, even in this setting (especially in the long-horizon scenario), bidirectional DTs may still suffer from the LDW problem, and decoding that is grounded in the adjacent context from both sides (start point and goal point) would be particularly advantageous in this setting. It becomes nearly impossible to effectively ground when the trajectory involves multiple intermediate waypoints.  In this scenario, Conv can be a better option than a more rigid semi-AR.
>
> ---
>
> References
>
> [1] Guanghan Wang et al., Remasking Discrete Diffusion Models with Inference-Time Scaling, 2025 arxiv
>
> [2] Lil Chen et al., Decision Transformer: Reinforcement Learning via Sequence Modeling NeurIPS 2021

---

### Decision · Program_Chairs · 2025-09-17

**Decision:**

Accept (spotlight)

**Comment:**

This paper addresses the long decoding-window (LDW) problem in diffusion-based language models, where tokens generated far from input context become repetitive or irrelevant. The authors propose two main contributions: Convolutional Decoding (Conv), which narrows the decoding window without block-wise segmentation while preserving bidirectionality and speed, and Rejecting Rule-based Fine-Tuning (R2FT), a post-hoc training method that reduces repetitive and high-prior token generation. The strengths include a clear problem formulation of an important limitation in diffusion LMs, novel and effective solutions that achieve state-of-the-art performance among diffusion models, comprehensive experimental evaluation on multiple benchmarks (AlpacaEval, MT-Bench, GSM8K), and strong theoretical grounding with hazard function analysis. The main weaknesses center on density and clarity issues that limit readability, limited evaluation of bidirectional tasks despite bidirectionality being a central claim, insufficient analysis of failure cases, and concerns about the generalizability of the rule-based R2FT approach beyond Q&A-like tasks.

The rebuttal process was notably thorough and constructive, with authors providing detailed theoretical explanations including hazard function analysis for Conv's advantages over semi-AR approaches, additional experimental results on reasoning tasks (GSM8K, MMLU) showing Conv's effectiveness, and clarification that unidirectional sampling was used exclusively in experiments. Reviewer QCLw increased their score and expressed strong support after the theoretical explanations, while Reviewer TLks raised important points about independence/irrevocability assumptions and copula-based alternatives, ultimately increasing their score to 5 after the discussion. Reviewer krtD initially had concerns about bidirectional claims being misleading but was satisfied with the authors' clarification about the distinction between bidirectional attention and bidirectional generation. The authors committed to improving clarity, toning down bidirectional claims, and incorporating all discussions into the camera-ready version. The overall reviewer sentiment shifted positively, with multiple score increases, strong endorsements from Reviewers QCLw and krtD, and constructive engagement from all reviewers, leading to a technically solid paper that advances diffusion LM capabilities despite some presentational issues.